# LO-SDA: Latent Optimization for Score-based Atmospheric Data Assimilation

## Abstract

Data assimilation (DA) plays a pivotal role in numerical weather prediction by systematically integrating sparse observations with model forecasts to estimate optimal atmospheric initial conditions for forthcoming forecasts. Traditional Bayesian DA methods adopt a Gaussian background prior as a practical compromise for the curse of dimensionality in atmospheric systems, which simplifies the nonlinear nature of atmospheric dynamics and can result in biased estimates. To address this limitation, we propose a novel generative DA method, LO-SDA. First, a variational autoencoder is trained to learn compact latent representations that disentangle complex atmospheric correlations. Within this latent space, a background-conditioned diffusion model is employed to directly learn the conditional distribution from data, thereby generalizing and removing assumptions in the Gaussian prior in traditional DA methods. Most importantly, we employ latent optimization during the reverse process of the diffusion model to ensure strict consistency between the generated states and sparse observations. Idealized experiments demonstrate that LO-SDA not only outperforms score-based DA methods based on diffusion posterior sampling but also surpasses traditional DA approaches. To our knowledge, this is the first time that a diffusion-based DA method demonstrates the potential to outperform traditional approaches on high-dimensional global atmospheric systems. These findings suggest that long-standing reliance on Gaussian priors—a foundational assumption in operational atmospheric DA—may no longer be necessary in light of advances in generative modeling.

## 1 Introduction

In numerical weather prediction, data assimilation (DA) is essential for generating accurate initial conditions that directly determine forecast skill (Lorenc, 1986; Gustafsson et al., 2018; Asch et al., 2016). Modern DA methods estimate the optimal atmospheric state $\boldsymbol{x}$ within a Bayesian framework by combining sparse observations $\boldsymbol{y}$ with model forecasts $\boldsymbol{x}_b$ (also known as background fields) (Asch et al., 2016; Rabier & Liu, 2003; Carrassi et al., 2018; Le Dimet & Talagrand, 1986). Specifically, DA aims to estimate the Bayesian posterior distribution $p(\boldsymbol{x}|\boldsymbol{x}_b, \boldsymbol{y})$. Given that forecasts and observations are typically conditionally independent, the posterior simplifies to $p(\boldsymbol{x}|\boldsymbol{x}_b, \boldsymbol{y}) \propto p(\boldsymbol{y}|\boldsymbol{x})p(\boldsymbol{x}|\boldsymbol{x}_b)$.

Traditional DA methods typically assume both the prior $p(\boldsymbol{x} \mid \boldsymbol{x}_b)$ and the likelihood $p(\boldsymbol{y}|\boldsymbol{x})$ follow Gaussian distributions to simplify the inference process (Bannister, 2017). While this assumption is relatively reasonable for observation errors, it breaks down for background uncertainty, which often becomes non-Gaussian after undergoing nonlinear model evolution. Furthermore, this assumption makes traditional DA methods rely on the background error covariance matrix $\mathbf{B}$ to define the solution space for assimilation (Bannister, 2008a). Nevertheless, $\mathbf{B}$ often spans more than $10^{12}$ degrees of freedom in high-resolution systems, making it extremely challenging to estimate and potentially introducing significant additional error into the assimilation process (Bannister, 2017; 2008b). These limitations have spurred generative DA frameworks.

Generative DA models perform posterior inference using score functions, offering a promising alternative to traditional approaches by relaxing the need for Gaussian assumptions (Rozet & Louppe, 2023b;a; Qu et al., 2024; Huang et al., 2024; Manshausen et al., 2025). However, existing approaches face notable limitations, both in practical implementation and theoretical understanding. For instance, DiffDA (Huang et al., 2024) conditions the diffusion model on the background and incorporates

observations through a repainting strategy, but their method underperforms in sparse observation settings and cannot effectively handle nonlinear observation operators such as satellite radiative transfer. Qu et al. (2024) encode background and multi-modal observations into a unified guidance signal, though their reliance on specific observation distribution assumptions restricts generalization to complex DA scenarios. Moreover, Rozet & Louppe (2023b;a) and Manshausen et al. (2025) treat observations as guidance during the reverse process, ignoring the background prior. While these work well when observations are dense and clean, they often fail under sparse or noisy conditions, where background information becomes essential. These limitations underscore the necessity of a unified framework that jointly leverages both background information and observational guidance in generative DA.

To this end, we propose the Latent Optimization Score-based Data Assimilation (LO-SDA) framework, which seeks to establish a novel bridge between score-based models and the principles of variational DA for a more fundamental and reliable formulation of generative DA. First, we train a variational autoencoder (VAE) to learn a compact latent representation of the high-dimensional atmospheric states, capturing nonlinear dependencies among variables and enabling more efficient probabilistic modeling. Second, we train a score-based model to learn the background conditioned prior in latent space $p(\boldsymbol{z}|\boldsymbol{z}_b)$, where $\boldsymbol{z}$ represents the latent representation of model state $\boldsymbol{x}$. Third, inspired by recent work on inverse problems in diffusion models (Song et al., 2024), we adapt its alternating latent optimization scheme to the context of Data Assimilation. This strategy iteratively enforces observational constraints during guided diffusion sampling. In our framework, the diffusion-estimated prior offers a more expressive and less biased analysis than traditional Gaussian assumptions (Figure 1 (a)). Critically, we employ an iterative latent optimization scheme that enforces strict analysis-observation consistency. As illustrated in Figure 1 (b), this iterative process distinguishes our work from single-step guidance methods like DPS. Each optimization step can be viewed as an attempt to minimize the observation-error cost. This multi-step maximization of the posterior likelihood is the key mechanism shared with variational DA that explains its superior performance. Furthermore, we demonstrate that LO-SDA is computationally efficient ( 1 minute per analysis) and stable for year-long sequential assimilation cycles (see Appendix).

Our contributions are outlined as follows:

- We establish a theoretical connection between latent optimization and variational DA, reformulating the iterative process as a generative analogue to traditional cost function minimization.

- We demonstrate for the first time that a score-based DA framework can surpass traditional DA approaches, and even match a state-of-the-art latent variational method (L3DVAR) on a high-dimensional global assimilation task.

- By iteratively enforcing observational constraints, our method produces more accurate and consistent analyses than common single-step guidance approaches and traditional DA.

## 2 RELATED WORK

**The variational assimilation methods.** Variational assimilation is a representative class of traditional DA methods and is widely used in operational numerical weather prediction systems. In the three-dimensional case, it seeks to maximize the posterior likelihood (Asch et al., 2016; Rabier & Liu, 2003; Carrassi et al., 2018):

$$\boldsymbol{x}_a = \arg\max_{\boldsymbol{x}} p(\boldsymbol{x}|\boldsymbol{x}_b, \boldsymbol{y}) = \arg\max_{\boldsymbol{x}} p(\boldsymbol{x}|\boldsymbol{x}_b)p(\boldsymbol{y}|\boldsymbol{x}), \tag{1}$$

where the assumption of independence between observation errors and background errors is applied. By assuming that the prior distribution $p(\boldsymbol{x}|\boldsymbol{x}_b)$ and the observation likelihood $p(\boldsymbol{y}|\boldsymbol{x})$ follow Gaussian distributions, three-dimensional variational DA (3DVar) is equivalent to minimizing the following cost function:

$$J(\boldsymbol{x}) = \frac{1}{2}(\boldsymbol{x} - \boldsymbol{x}_b)^T \mathbf{B}^{-1}(\boldsymbol{x} - \boldsymbol{x}_b) + \frac{1}{2}(\boldsymbol{y} - \mathcal{H}(\boldsymbol{x}))^T \mathbf{R}^{-1}(\boldsymbol{y} - \mathcal{H}(\boldsymbol{x})) . \tag{2}$$

where $\mathbf{B}$ and $\mathbf{R}$ denote the covariance matrices of background and observation errors, respectively, and $\mathcal{H}$ is the observation operator that maps model states to observation space. As noted by Bannister

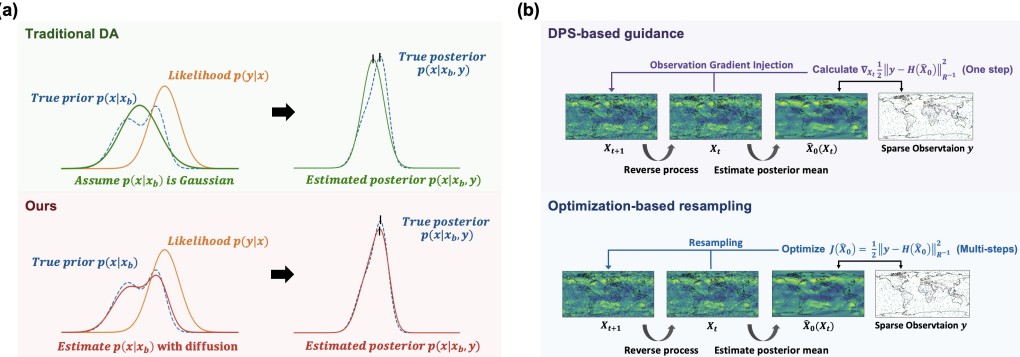

Figure 1: Comparison between LOSDA and other DA approaches. (a) Prior estimation: The true background conditional prior $p(\boldsymbol{x}|\boldsymbol{x}_b)$ (blue dashed) is approximated as Gaussian in traditional DA (green), while LOSDA directly estimates it through diffusion modeling (red). By incorporating observation likelihood $p(\boldsymbol{y}|\boldsymbol{x})$, LOSDA achieves posterior estimation $p(\boldsymbol{x}|\boldsymbol{x}_b, \boldsymbol{y})$ closer to the ground truth. (b) Observation integration methods: Top - Diffusion Posterior Sampling (DPS) updates denoised $\boldsymbol{x}_t$ via observation error gradient guidance (single-step consistency). Bottom - LOSDA's optimization approach directly minimizes observation error for optimal denoised $\boldsymbol{x}_t$ (strict multi-step consistency). Our framework enforces tighter observation constraints than gradient-based DPS.

(2008a; 2017), $\mathbf{B}$ plays a central role in variational DA by shaping the feasible solution space and promoting physical consistency in the resulting analysis. In practice, the high-dimensional $\mathbf{B}$ is commonly simplified via a control variable transformation that approximately diagonalizes it (Descombes et al., 2015). Although this facilitates its inversion, the simplified $\mathbf{B}$ may fail to capture the evolving physical consistency of atmospheric states, leading to suboptimal assimilation outcomes.

**The latent assimilation methods.** Recently, latent data assimilation (LDA) (Cheng et al., 2024; Melinc & Zaplotnik, 2024; Peyron et al., 2021; Amendola et al., 2021; Zheng et al.; Fan et al., 2025a;b;c) has been proposed to apply traditional DA methods with Gaussian priors in a compact latent space learned via autoencoders. For example, the latent formulation of the widely used 3DVar—referred to as L3DVar—optimizes the following loss function:

$$ J(\boldsymbol{z}) = \frac{1}{2}(\boldsymbol{z} - \boldsymbol{z}_b)^T \mathbf{B}_z^{-1}(\boldsymbol{z} - \boldsymbol{z}_b) + \frac{1}{2}(\boldsymbol{y} - \mathcal{H}(D(\boldsymbol{z}))^T \mathbf{R}^{-1}(\boldsymbol{y} - \mathcal{H}(D(\boldsymbol{z}))). \qquad (3) $$

where $\boldsymbol{z}$ and $\mathbf{B}_z$ denote the latent state and the background error covariance matrix in the latent space, respectively. Several studies have found that $\mathbf{B}_z$ is inherently near-diagonal, as the latent space effectively captures correlations among atmospheric variables. Consequently, LDA can adopt a diagonal $\mathbf{B}_z$, greatly simplifying its implementation (Melinc & Zaplotnik, 2024; Zheng et al.; Fan et al., 2025a). Fan et al. (2025a) further showed that performing variational assimilation in latent space can outperform its model-space counterpart. Nevertheless, most latent DA methods remains constrained by the Gaussian prior assumption. To overcome this limitation, our work also leverages latent representations of high-dimensional atmospheric states, but replaces the Gaussian prior with a more expressive, data-driven distribution modeled by a latent score-based model.

**Generative DA.** Recent works have explored integrating generative models with DA. unlike Latent-EnSF Si & Chen (2025) and LD-EnSF Anonymous (2025) which apply ensemble score filters in latent space, LO-SDA employs a single background-conditioned diffusion prior, avoiding the complexity of ensemble simulations. Compared to methods like APPA Andry et al. (2025) or other SDA Huang et al. (2024); Manshausen et al. (2025); Rozet & Louppe (2023b); Qu et al. (2024) that use unconditional priors or single-step guidance, LO-SDA's iterative optimization serves as a generative analogue to the variational cost minimization, enabling superior performance in high-dimensional, sparse-observation regimes.

## 3 METHOD

### 3.1 PRELIMINARY

**Score-based model.** The score-based model, a promising class of the generative models (Dhariwal & Nichol, 2024; Ho et al., 2022), offering high-quality generation and excellent model convergence (Song et al., 2021a; Huang et al., 2021; Kingma et al., 2021). It comprises a forward process and a reverse process (Ho et al., 2020; Sohl-Dickstein et al., 2015; Song et al., 2021b). In the forward process, the original data distribution is transformed into a known prior, by gradually injecting noise. Such a process is governed by a stochastic differential equation (SDE) and a corresponding reverse-time SDE (Song et al., 2021b),

$$d\boldsymbol{x} = \mathbf{f}(\boldsymbol{x}, t)dt + g(t)d\boldsymbol{w} \tag{4}$$

$$d\boldsymbol{x} = [\mathbf{f}(\boldsymbol{x}, t) - g(t)^2 \nabla_{\boldsymbol{x}} \log p_t(\boldsymbol{x})]dt + g(t)d\bar{\boldsymbol{w}}, \tag{5}$$

where the reverse SDE transforms the prior distribution back into the data distribution by gradually removing the noise. Here, $\boldsymbol{w}$ and $\bar{\boldsymbol{w}}$ both represent the standard Wiener processes (Gaussian white noise), with $\mathbf{f}(\boldsymbol{x}, t)$ the drift coefficient and $g(t)$ the diffusion coefficient of $\boldsymbol{x}(t)$. Accordingly, the perturbation kernel from $\boldsymbol{x}_0$ to $\boldsymbol{x}_t$ takes form $p(\boldsymbol{x}_t|\boldsymbol{x}) \sim \mathcal{N}(\mu(t), \sigma^2(t)\boldsymbol{I})$, where $\mu(t), \sigma^2(t)$ can be determined by the $\mathbf{f}(\boldsymbol{x}, t)$ and $g(t)$. In this work, we take the widely used variance-preserving SDE and the cosine schedule for $\mu(t)$ (Rozet & Louppe, 2023b). In the generative diffusion model, the score function $\nabla_{\boldsymbol{x}} \log p_t(\boldsymbol{x})$ can be estimated by a neural network with parameter $\boldsymbol{\theta}$ via minimizing the denoising score matching loss $\mathcal{L}_t \equiv \mathbb{E}_{p(\boldsymbol{x}_t)}||\boldsymbol{s}_{\boldsymbol{\theta}}(\boldsymbol{x}, t) - \nabla_{\boldsymbol{x}} \log p_t(\boldsymbol{x}|\boldsymbol{x}_0)||^2$, which theoretically guarantees $\boldsymbol{s}_{\boldsymbol{\theta}}(\boldsymbol{x}, t) \approx \nabla_{\boldsymbol{x}} \log p_t(\boldsymbol{x})$ (Song et al., 2021b). Once we have a trained $s_\theta(\boldsymbol{x}, t)$, the trajectory from the prior distribution to the real data distribution can be determined following Equation 5.

**Score-based data assimilation.** Data assimilation under this framework reformulates the Bayesian posterior as a composite scoring process:

$$\nabla_{\boldsymbol{x}_t} \log p(\boldsymbol{x}_t|\boldsymbol{x}_b, \boldsymbol{y}) = \nabla_{\boldsymbol{x}_t} \log p(\boldsymbol{x}|\boldsymbol{x}_b) + \nabla_{\boldsymbol{x}_\tau} \log p(\boldsymbol{y}|\boldsymbol{x}_t) \tag{6}$$

The prior term leverages the diffusion model's capacity to capture complex spatial correlations, bypassing the oversimplified Gaussian assumptions in the conventional DA methods. The constraint term enforces observation consistency through conditional guidance. In the DPS paradigm (Chung et al., 2023), the observation term is supposed to follow Gaussian distribution,

$$p(\boldsymbol{y}|\boldsymbol{x}_t) \sim \mathcal{N}\left(\mathcal{H}(\tilde{\boldsymbol{x}}_0(\boldsymbol{x}_t)), \boldsymbol{R}\right) \tag{7}$$

where the posterior mean $\tilde{\boldsymbol{x}}_0$ derives from Tweedie's formula (Ho et al., 2020; Sohl-Dickstein et al., 2015):

$$\tilde{\boldsymbol{x}}_0(\boldsymbol{x}_t) = \frac{\boldsymbol{x}_t + \sigma(t)^2 \boldsymbol{s}_{\boldsymbol{\theta}}(\boldsymbol{x}_t, \boldsymbol{x}_b)}{\mu(t)}. \tag{8}$$

### 3.2 LATENT SCORE-BASED DATA ASSIMILATION

Due to the computational challenges in high-dimensional systems, LDA (Cheng et al., 2024; Melinc & Zaplotnik, 2024; Peyron et al., 2021; Amendola et al., 2021; Fan et al., 2025a) is proposed to leverage VAEs for compressing physical fields into low-dimensional manifolds (Kingma et al., 2013; Doersch, 2016) and performing efficient optimization in this reduced space. Specifically, the traditional 3DVar formulation (Equation 1) is adapted to the latent space with cost function described by Equation 3. The gradient descent iteratively optimizes the latent representation of the analysis field. The optimized latent is then decoded to reconstruct the assimilated state. Although LDA alleviates non-linearity challenges, its retention of Gaussian assumptions for latent background distributions imposes theoretical limitations as above-discussed, particularly in capturing the multiscale complexity characteristic of real atmospheric states (Fan et al., 2025a). In this work, we train a score-based model in the latent space to model the background conditional distribution. Additionally, we integrate observations through guidance sampling in the latent space. Mathematically, the latent score modeling for DA can be expressed as:

$$\nabla_{\boldsymbol{z}_t} \log p(\boldsymbol{z}_t|\boldsymbol{z}_b, \boldsymbol{y}) = \nabla_{\boldsymbol{z}_t} \log p(\boldsymbol{z}|\boldsymbol{z}_b) + \nabla_{\boldsymbol{z}_t} \log p(\boldsymbol{y}|\boldsymbol{z}_t)$$
$$= \boldsymbol{s}_{\boldsymbol{\theta}}(\boldsymbol{z}_t, \boldsymbol{z}_b) + \nabla_{\boldsymbol{z}_t} \log p(\boldsymbol{y}|\boldsymbol{z}_t). \tag{11}$$

---

**Algorithm 1** Comparison of DPS Guidance and Latent Optimization for Score-Based Data Assimilation

---

1: **Input:** Pretrained score function $s_{\boldsymbol{\theta}}(\boldsymbol{z}_t, \boldsymbol{z}_b) = \nabla_{\boldsymbol{z}_t} \log p(\boldsymbol{z}_t|\boldsymbol{z}_b)$, pretrained VAE (encoder $E(\cdot)$, decoder $D(\cdot)$), observation distribution $p(\boldsymbol{y}|\boldsymbol{z}_t)$, observations $\boldsymbol{y}$.
2: **for** $t = 1$ to $0$ **do**
3:    Solve reverse SDE with $\boldsymbol{z}_{t+1}$ and score function $\nabla_{\boldsymbol{z}_t} p(\boldsymbol{z}_t|\boldsymbol{z}_b)$: $\tilde{\boldsymbol{z}}_t \leftarrow \text{SolutionAtTime}(t)$
4:    **if** $t \in C$ **then**
5:       Calculate posterior mean: $\hat{\boldsymbol{z}}_0 = \frac{\tilde{\boldsymbol{z}}_t + \sigma^2(t)\nabla_{\boldsymbol{z}_t} \log p(\boldsymbol{z}_t|\boldsymbol{z}_b)}{\mu(t)}$

6:       **DPS guidance**
7:       Perform diffusion posterior sampling:

$$\boldsymbol{z}_t = \tilde{\boldsymbol{z}}_t + \zeta\nabla_{\boldsymbol{z}_t} \log p(\boldsymbol{y}|\boldsymbol{z}_t)$$
$$= \tilde{\boldsymbol{z}}_t - \frac{1}{2}\zeta\nabla_{\tilde{\boldsymbol{z}}_t}(\boldsymbol{y} - \mathcal{H}(D(\hat{\boldsymbol{z}}_0)))^T \boldsymbol{R}^{-1}(\boldsymbol{y} - \mathcal{H}(D(\hat{\boldsymbol{z}}_0))) \qquad (9)$$

8:       **Latent Optimization**
9:       With initial value $\boldsymbol{z}^0 = \hat{\boldsymbol{z}}_0$
10:      **Repeat**

$$\boldsymbol{z}^{i+1} = \boldsymbol{z}^i + \zeta\nabla_{\boldsymbol{z}^i} \log p(\boldsymbol{y}|\boldsymbol{z}^i)$$
$$= \boldsymbol{z}^i - \frac{1}{2}\zeta\nabla_{\boldsymbol{z}^i}(\boldsymbol{y} - \mathcal{H}(D(\boldsymbol{z}^i)))^T \boldsymbol{R}^{-1}(\boldsymbol{y} - \mathcal{H}(D(\boldsymbol{z}^i))) \qquad (10)$$

11:      **Until** Convergence to $\hat{\boldsymbol{z}}_0(\boldsymbol{y})$
12:      Go back to the noising manifold by resampling: $\boldsymbol{z}_t \sim p(\boldsymbol{z}_t|\tilde{\boldsymbol{z}}_t, \hat{\boldsymbol{z}}_0(\boldsymbol{y}), \boldsymbol{y})$
13:    **else**
14:       $\boldsymbol{z}_t = \tilde{\boldsymbol{z}}_t$
15:    **end if**
16: **end for**
17: **Return:** The decoded optimized latent variables $D(\boldsymbol{z}_0)$

---

For the guidance term, we implemented the latent counterpart of DPS guidance where the observation term preserves Gaussian distributions $p(\boldsymbol{y}|\boldsymbol{z}_t) \sim \mathcal{N}(\mathcal{H}(D(\hat{\boldsymbol{z}}_0(\boldsymbol{z}_t))), \boldsymbol{R})$. Similar to Equation 8, $\hat{\boldsymbol{z}}_0$ is the posterior mean. $D(\cdot)$ denotes the decoder of VAE. While Algorithm 1 outlines the sampling process using DPS guidance, its single-step gradient update mechanism may provide insufficient constraint enforcement, potentially compromising observation consistency in high-dimensional scenarios.

### 3.3 LATENT OPTIMIZATION TECHNIQUES

To perform strict observation consistency, we aim to integrate variational optimization (Equation 2 and Equation 3) used in traditional DA. Inspired by inverse problem solving techniques (Song et al., 2024) within diffusion models, a two-stage latent optimization strategy is proposed: Hard-Constrained Optimization: (1) Solving $\hat{\boldsymbol{z}}_0(\boldsymbol{y}) = \arg\min_{\boldsymbol{z}}(\boldsymbol{y} - \mathcal{H}(D(\boldsymbol{z})))^T \boldsymbol{R}^{-1}(\boldsymbol{y} - \mathcal{H}(D(\boldsymbol{z})))$ to ensure strict observation consistency, (2) Projecting the optimized latent back to the noisy data manifold using the reverse process. Since the $\hat{\boldsymbol{z}}_0(\boldsymbol{y}) = (\boldsymbol{z}_t - \sigma(t)\varepsilon)/\mu(t)$ can be viewed as the estimated mean of latent $\boldsymbol{z}_t$ with the observation single $\boldsymbol{y}$, one have that

$$p(\boldsymbol{z}_t|\hat{\boldsymbol{z}}_0(\boldsymbol{y}), \boldsymbol{y}) \sim \mathcal{N}\left(\mu(t)\hat{\boldsymbol{z}}_0(\boldsymbol{y}), \sigma^2(t)\boldsymbol{I}\right), \qquad (12)$$

from the forward process. When we map the $\hat{\boldsymbol{z}}_0(\boldsymbol{y})$ back to noise data manifold, we need the distributions $p(\boldsymbol{z}_t|\tilde{\boldsymbol{z}}_t, \hat{\boldsymbol{z}}_0(\boldsymbol{y}), \boldsymbol{y})$. By Bayesian formula, $p(\boldsymbol{z}_t|\tilde{\boldsymbol{z}}_t, \hat{\boldsymbol{z}}_0(\boldsymbol{y}), \boldsymbol{y}) \propto p(\tilde{\boldsymbol{z}}_t|\boldsymbol{z}_t, \hat{\boldsymbol{z}}_0(\boldsymbol{y}), \boldsymbol{y})p(\boldsymbol{z}_t|\hat{\boldsymbol{z}}_0(\boldsymbol{y}), \boldsymbol{y})$. The posterior distribution $p(\tilde{\boldsymbol{z}}_t|\boldsymbol{z}_t, \hat{\boldsymbol{z}}_0(\boldsymbol{y}), \boldsymbol{y})$ is assumed as Gaussian distribution with variance $\lambda_t^2$ and the $p(\boldsymbol{z}_t|\hat{\boldsymbol{z}}_0(\boldsymbol{y}), \boldsymbol{y})$ is supposed to provide the prior of its mean. Thus, it is accordingly derived (see Appendix):

$$p(\boldsymbol{z}_t|\tilde{\boldsymbol{z}}_t, \hat{\boldsymbol{z}}_0(\boldsymbol{y}), \boldsymbol{y}) = \mathcal{N}\left(\frac{\lambda_t^2\mu(t)\hat{\boldsymbol{z}}_0(\boldsymbol{y}) + \sigma^2(t)\tilde{\boldsymbol{z}}_t}{\lambda_t^2 + \sigma^2(t)}, \frac{\lambda_t^2\sigma^2(t)}{\lambda_t^2 + \sigma^2(t)}\boldsymbol{I}\right). \qquad (13)$$

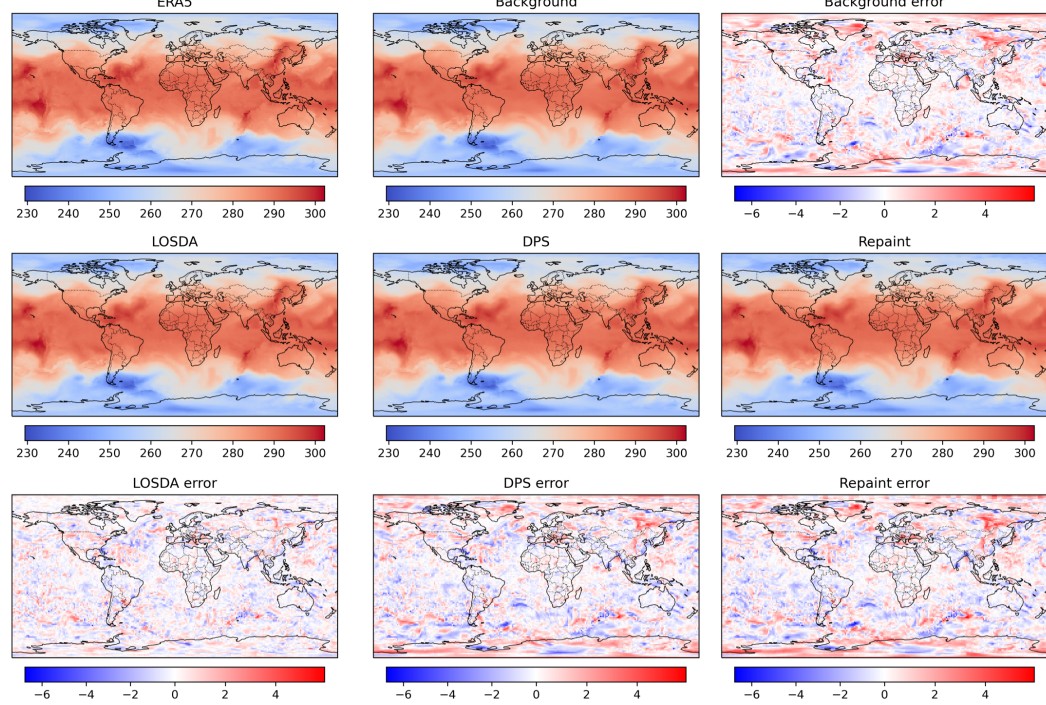

Figure 2: Comparative visualization of t850 analysis fields across assimilation methods under 1% idealized observation (valid at 2019-01-03 00:00 UTC). Top row (left to right): ERA5 ground truth, background field, and background error. Middle row: Assimilation results from (a) proposed LO-SDA method, (b) DPS framework, and (c) Repaint approach. Bottom row: Corresponding absolute error fields relative to ERA5 truth. The reduced error magnitude (lighter hues) in LO-SDA results demonstrates our method's superior error reduction capability compared to alternative approaches.

Following Song et al. (2024), we choose the variance $\lambda_t^2$ schedule as $\lambda_t^2 = \lambda \left( \frac{1-\mu^2(t-\Delta t)}{\mu^2(t)} \right) \left( 1 - \frac{\mu^2(t)}{\mu^2(t-\Delta t)} \right)$ with a hyperparameter $\lambda$. This approach integrates variational optimization within the diffusion sampling framework (Algorithm 1), where latent variables are iteratively refined at multiple diffusion steps, preserving highly observation consistency.

We now formalize the theoretical connection between the latent optimization technique and the principles of variational DA. The analysis field in variational DA is obtained by minimizing a cost function (Equation 2) that balances a background term against an observation term. Instead of enforcing an **approximated Gaussian prior** via a term like $(\boldsymbol{z} - \boldsymbol{z}_b)^\top \boldsymbol{B}_z^{-1}(\boldsymbol{z} - \boldsymbol{z}_b)$, our framework leverages the diffusion model itself as a powerful, data-driven **generative prior**, which serves the role of the background term. For the observation term, in contrast to single-step guidance methods like DPS, our latent optimization step (Equation 10) **iteratively seeks a state** $\hat{\boldsymbol{z}}_0(\boldsymbol{y})$ that minimizes the observation-error cost, $\|\boldsymbol{y} - \mathcal{H}(\mathcal{D}(\boldsymbol{z}^i))\|^2$. This objective is functionally equivalent to the observation term in the variational cost function, explicitly pulling the solution towards **strict data consistency**. The subsequent **resampling step** (Line 12 in Algorithm 1), $\boldsymbol{z}_t' \sim p(\boldsymbol{z}_t | \tilde{\boldsymbol{z}}_t, \hat{\boldsymbol{z}}_0(\boldsymbol{y}), \boldsymbol{y})$ This step projects the observation-consistent state back onto the complex, non-Gaussian manifold learned by the model, ensuring physical plausibility.

Accordingly, the LO-SDA framework effectively splits the variational optimization into an iterative procedure of alternating optimization (for the observation term) and projection (for the background term). This deep, optimization-based similarity not only motivates our approach but also provides a compelling explanation for its outstanding performance, as it marries the rigorous observation constraint of variational methods with the expressive power of a learned, non-Gaussian prior.

## 4 EXPERIMENTS

### 4.1 EXPERIMENTAL SETTINGS AND EVALUATIONS

**Dataset and metrics.** We conduct our experiments on the ERA5 reanalysis dataset (Hersbach et al., 2020), a global atmospheric data product maintained by the European Centre for Medium-Range

Weather Forecasts (ECMWF). Our study utilizes 5 upper-air atmospheric variables (geopotential, temperature, specific humidity, zonal wind, and meridional wind) across 13 pressure levels (50hPa, 100hPa, 150hPa, 200hPa, 250hPa, 300hPa, 400hPa, 500hPa, 600hPa, 700hPa, 850hPa, 925hPa, and 1000hPa), combined with 4 surface variables (10-meter zonal component of wind (u10), 10-meter meridional component of wind (v10), 2-meter temperature (msl) and mean sea level pressure (msl)), forming a total of 69 meteorological variables. The pressure-level variables follow the standardized ERA5 naming convention (e.g., t850 denotes temperature at 850 hPa). We use a subset spanning 1979-2018 for training and evaluations. For evaluations, the assimilation quality is assessed by direct comparison with ERA5 reference fields. Three metrics quantifying performance are overall mean square error (MSE), mean absolute error (MAE), and the latitude-weighted root mean square error (WRMSE) (see Appendix), which is a statistical metric widely used in geospatial analysis and atmospheric science (Rasp et al., 2020; 2024). The validation procedure conducts assimilation cycles at 00:00 UTC for each day throughout 2019. For each test case, we calculate the above three metrics. Final performance scores represent the annual average of these daily metrics, ensuring statistically significant results across all seasons and synoptic conditions.

**Experimental setting.** The Fengwu AI forecasting model (Chen et al., 2023) (6-hour temporal resolution) is integrated into our DA framework to produce the background field. These fields are generated through an 8-step autoregressive forecasting procedure, initialized with ERA5 conditions from 48 hours prior to the target assimilation lead time. To simulate realistic observing system characteristics, we create synthetic observations by randomly masking the ERA5 truth data at two sparsity levels (95% and 99%), mimicking typical satellite coverage constraints. The 1.40625° ($128 \times 256$ grid) spatial resolution is employed, yielding input arrays of size $69 \times 128 \times 256$.

**The background conditional diffusion model.** We present a unified framework for conditional physics field modeling through variational autoencoding and latent diffusion. Our architecture begins with a window-attention transformer VAE (Han et al., 2024) that compresses high-dimensional fields ($69 \times 128 \times 256$) to compact latent representations ($69 \times 32 \times 64$). Trained for 80 epochs using AdamW (Loshchilov et al., 2017) with batch size 32, the VAE employs a hybrid learning rate schedule: linear warmup to $2 \times 10^{-4}$ over 10,000 iterations followed by cosine decay, achieving 0.0067 reconstruction MSE as detailed in Appendix. The latent diffusion process then learns conditional distributions $p(\boldsymbol{z}|\boldsymbol{z}_b)$ through a 28-layer transformer backbone (Peebles & Xie, 2023) with 1152-dimensional hidden states, (2,2) patch embedding, and 16-head cross-attention for background latent $\boldsymbol{z}_b$ conditioning. For diffusion setting, the variance-preserving SDE (Song et al., 2021b) with cosine noise scheduling. Optimized via AdamW (Loshchilov et al., 2017) at constant $1 \times 10^{-4}$ learning rate (batch size 32), the model converges stably over 100k training steps. Sampling employs a modified Predictor-Corrector scheme combining 128-step prediction with 2 iterations of Langevin correction (Song et al., 2021b). See the Appendix for more comprehensive resource usage.

**Baselines.** For diffusion-based experiments, we incorporate observations through two baseline methods: a latent-space implementation of the repainting technique from DiffDA (Huang et al., 2024) and the latent version of DPS described in Algorithm 1. The latent repaint implementation follows:

$$\boldsymbol{z}_t^{obs} \sim \mathcal{N}(\mu(t)E(\boldsymbol{x}^*), \sigma^2(t)\boldsymbol{I}), \quad \tilde{\boldsymbol{z}}^t \leftarrow \text{SolutionAtTime}(t) \tag{14}$$

$$\boldsymbol{z}_{t-1} = E\left(m \odot D(\boldsymbol{z}_t^{obs}) + (1-m) \odot D(\tilde{\boldsymbol{z}}^t)\right) \tag{15}$$

where $z_t^{obs}$ is noised latent with $E(\boldsymbol{x}^*)$ is the encoded ERA5 ground truth latent. $\boldsymbol{z}_t$ is the sampled prior latent at diffusion time $t$. We finally combine the decoded observed and prior latents in model space using a masking matrix $m$, with $\odot$ denoting element-wise multiplication. Moreover, we compare against not only the conventional 3DVar but also its powerful latent-space counterpart, L3DVAR, which represents the state-of-the-art for **machine learning based variational methods**.

## 4.2 RESULTS

Table 1 presents our key results (see Appendix for statistical significance analysis). ~~, which demonstrate a notable improvement for generative data assimilation.~~ We note that Repaint performs poorly in the 1% observation setting. This may be because its masking/inpainting mechanism fails to effectively propagate information from extremely sparse points to the global field, leading to artifacts in unobserved regions. Nevertheless, when compared to traditional 3DVAR data assimilation, LO-SDA exhibits significantly improved accuracy. The most critical comparison is against L3DVAR,

Table 1: Quantitative performance comparison of different methods under 1% and 5% observations. The red and yellow indicate the first and second best performing methods.

| Ratio | Model | MSE | MAE | WRMSE | | | | | |
| | | | | msl | u10 | u700 | v500 | z500 | t850 |
|---|---|---|---|---|---|---|---|---|---|
| | 48h background | 0.0505 | 0.1178 | 98.7265 | 1.2727 | 1.9953 | 2.4217 | 89.2752 | 0.9310 |
| 1% observation | 3DVAR | 0.0483 | 0.1138 | 81.6384 | 1.2235 | 1.9850 | 2.4298 | 62.9377 | 0.8797 |
| | L3DVAR | 0.0474 | 0.1105 | 62.1054 | 1.1862 | 1.9797 | 2.3392 | 53.0902 | 0.8975 |
| | Repaint | 0.0592 | 0.1311 | 114.3672 | 1.4167 | 2.1059 | 2.5664 | 104.1351 | 1.0363 |
| | DPS | 0.0545 | 0.1247 | 95.9850 | 1.3286 | 2.0220 | 2.3981 | 85.5673 | 1.0269 |
| | LO-SDA(ours) | 0.0472 | 0.1101 | 62.4505 | 1.1836 | 1.8981 | 2.2439 | 53.6468 | 0.9243 |
| 5% observation | 3DVAR | 0.0430 | 0.0982 | 63.2562 | 1.1661 | 1.8765 | 2.2365 | 45.8574 | 0.7849 |
| | L3DVAR | 0.0315 | 0.0903 | 45.8037 | 0.9350 | 1.7166 | 1.9400 | 38.6411 | 0.8024 |
| | Repaint | 0.0496 | 0.1219 | 106.3938 | 1.2934 | 1.9889 | 2.3622 | 95.9167 | 0.9856 |
| | DPS | 0.0486 | 0.1199 | 93.2470 | 1.2673 | 1.9585 | 2.3271 | 85.3329 | 0.9891 |
| | LO-SDA(ours) | 0.0309 | 0.0851 | 42.3498 | 0.8992 | 1.5873 | 1.7894 | 32.8990 | 0.8094 |

a state-of-the-art, machine learning-driven variational method. Even under the highly sparse 1% observation setting, LO-SDA achieves performance that is statistically comparable to this highly optimized baseline. This result demonstrates, for the first time, that a score-based framework can achieve comparable performance to SOTA machine learning driven variational DA (L3DVAR) at the global atmospheric scale. Furthermore, this performance advantage widens as observational density increases. At 5% observation coverage, LO-SDA decisively surpasses L3DVAR, delivering a **14.86%** improvement for z500 WRMSE. This superior scalability highlights the effectiveness of our approach in leveraging denser observational constraints, a key requirement for operational systems. Beyond the idealized settings, we evaluate LO-SDA with a real-world observation distribution of approximately 0.6% coverage (see Appendix for more details). This evaluation demonstrates that our framework's performance is comparable to L3DVAR even under highly sparse, realistic conditions.

This success can be attributed to the **generative variational principles** in our framework, which fundamentally distinguish LO-SDA from other diffusion-based approaches. While methods like DPS rely on a single-step gradient correction, LO-SDA's iterative optimization enforces rigorous data consistency, as theoretically motivated in Algorithm 1. The advantage of this iterative strategy is further demonstrated by the comparison with other diffusion-based methods: under 1% observations, LO-SDA improves upon DPS and Repaint by **13.39%** and **20.27%** in overall MSE, respectively. This empirically validates our hypothesis that integrating variational-like optimization is key to unlocking the full potential of diffusion models for DA.

LO-SDA also demonstrates that generative modeling can transcend traditional Gaussian assumptions without compromising accuracy. This advancement stems from a novel hybrid approach: a background-conditioned diffusion model replaces the restrictive prior, while iterative latent optimization integrates the flexibility of generative modeling with the rigorous observation-consistency requirements of operational DA. These findings establish score-based models as a well-established and highly competitive paradigm for the future of DA.

### 4.3 ABLATION STUDIES

**Observation Error Robustness.** To evaluate the robustness of LO-SDA under realistic observational conditions, we conduct experiments with simulated observation errors by injecting additive Gaussian noise with standard deviations of 2%, 5%, and 10% of the ERA5 climatological standard deviation. As evidenced by the quantitative results in Table 2, our framework maintains consistent performance across various noise levels, demonstrating remarkable error tolerance. In fact, by tuning the optimization constraint strength, our method can flexibly balance between observation consistency and diffusion prior (see Appendix for detailed analysis).

**Latent Optimization Frequency Analysis.** To empirically validate the theoretical connection (see Section 3.3) and demonstrate its impact on our framework's effectiveness, we conduct an ablation study on the latent optimization frequency by adjusting the skip interval parameter. Table 3 reveals a systematic performance degradation as the skip interval increases (i.e., fewer optimization steps). Specifically, under sparse 5% observation conditions, the overall MSE increases by 19.74% and

Table 2: Quantitative performance comparison under a 1% observation setting, with varying observation errors modeled as Gaussian noise. The standard deviations are set to 0.02, 0.05, and 0.10 relative to the ERA5 climatological standard deviation. The red and yellow indicate the first and second best performing methods.

| Ratio | Model | MSE | MAE | WRMSE | | | | | |
|-------|-------|-----|-----|-------|----|----|----|----|----|
| | | | | msl | u10 | u700 | v500 | z500 | t850 |
| | 48h background | 0.0505 | 0.1178 | 98.7265 | 1.2727 | 1.9953 | 2.4217 | 89.2752 | 0.9310 |
| std = 0.02 | 3DVAR | 0.0484 | 0.1141 | 82.4252 | 1.2243 | 1.9805 | 2.4202 | 67.3522 | 0.8679 |
| | L3DVAR | 0.0475 | 0.1109 | 63.0360 | 1.1935 | 1.9900 | 2.3496 | 53.6535 | 0.9054 |
| | LO-SDA(ours) | 0.0470 | 0.1109 | 64.7142 | 1.1940 | 2.0618 | 2.2888 | 55.3858 | 0.9354 |
| std = 0.05 | 3DVAR | 0.0485 | 0.1158 | 85.4325 | 1.2246 | 1.9643 | 2.3893 | 77.8867 | 0.8845 |
| | L3DVAR | 0.0481 | 0.1132 | 72.2672 | 1.1852 | 1.9515 | 2.3199 | 63.6825 | 0.9143 |
| | LO-SDA(ours) | 0.0479 | 0.1127 | 68.0784 | 1.1989 | 1.9104 | 2.2641 | 58.1329 | 0.9542 |
| std = 0.10 | 3DVAR | 0.0495 | 0.1173 | 91.2331 | 1.2291 | 1.9493 | 2.3636 | 84.4765 | 0.9101 |
| | L3DVAR | 0.0489 | 0.1147 | 83.1325 | 1.1954 | 1.9448 | 2.3275 | 75.3260 | 0.9341 |
| | LO-SDA(ours) | 0.0498 | 0.1188 | 82.0772 | 1.2408 | 1.9602 | 2.3194 | 69.4214 | 1.0308 |

Table 3: Comparison of latent optimization frequencies (skip=2, 4, 8) in reverse diffusion sampling under sparse observation settings (1% and 5%).

| Ratio | Frequency | MSE | MAE | WRMSE | | | | | |
|-------|-----------|-----|-----|-------|----|----|----|----|----|
| | | | | msl | u10 | u700 | v500 | z500 | t850 |
| | 48h background | 0.0505 | 0.1178 | 98.7265 | 1.2727 | 1.9953 | 2.4217 | 89.2752 | 0.9310 |
| 1% observation | skip=2 | 0.0472 | 0.1101 | 62.4505 | 1.1836 | 1.8981 | 2.2439 | 53.6468 | 0.9243 |
| | skip=4 | 0.0518 | 0.1162 | 70.2005 | 1.2648 | 1.9677 | 2.33571 | 59.6587 | 0.9611 |
| | skip=8 | 0.0549 | 0.1205 | 74.2434 | 1.3258 | 2.0143 | 2.4029 | 62.7748 | 0.9840 |
| 5% observation | skip=2 | 0.0309 | 0.0851 | 42.3498 | 0.8992 | 1.5873 | 1.7894 | 32.8990 | 0.8094 |
| | skip=4 | 0.0370 | 0.0939 | 49.5966 | 0.9892 | 1.6697 | 1.8892 | 41.5379 | 0.8538 |
| | skip=8 | 0.0415 | 0.1001 | 54.9766 | 1.0740 | 1.7544 | 2.0039 | 45.8147 | 0.8890 |

34.30% for skip intervals of 4 and 8, respectively, compared to the baseline configuration with skip=2. This degradation aligns with our theoretical insight: frequent latent optimization ensures proper integration of observations into the diffusion process, analogous to how iterative refinement in variational DA minimizes the analysis cost function.

## 4.4 COMPUTATIONAL EFFICIENCY

While variational methods like 3DVAR and L3DVAR are highly efficient (seconds per analysis on GPU), generative approaches typically require more computational resources. To identify the bottleneck in our framework, we performed a detailed runtime breakdown analysis on a single NVIDIA A100 GPU. As shown in the Appendix, the iterative latent optimization is the dominant cost, consuming ∼90% of the total runtime. This confirms that the computational burden stems from the rigorous enforcement of observation constraints.

However, this cost is not fixed. Our convergence analysis (detailed in the Appendix) reveals that it is promising to reduce the total analysis time to about 1 minute with fewer diffusion sampling steps and iterations, making LO-SDA computationally competitive. Moreover, by leveraging established diffusion sampling acceleration techniques, it is promising to reduce this cost to seconds, moving towards the goal of real-time assimilation.

~~The variational methods like 3DVAR and L3DVAR (in the latent space) are highly efficient (5 and 10 seconds per assimilation on GPU, respectively). A standard diffusion sampler without optimization takes approximately 100 seconds. Table 3 reveals the trade-off between assimilation accuracy and computational efficiency. The cost increases linearly as the optimization becomes more frequent (i.e., as the skip interval decreases). The two-skip configuration requires about 5 minutes (∼300 seconds). While this represents a significant computational demand, LO-SDA is~~

~~the first diffusion-based framework that validates a paradigm shift from well-established variational techniques to more powerful, data-driven generative approaches.~~

## 5 CONCLUSION

The proposed LO-SDA framework presents a significant advancement in data assimilation by introducing a generative approach that effectively overcomes the limitations of traditional Gaussian assumptions. We reformulate the iterative latent optimization technique as a generative analogue to traditional cost function minimization, where an optimization step enforces observation consistency and a resampling step enforces a powerful, data-driven prior. The practical impact of this connection is significant: experimental results show that LO-SDA is the first score-based framework to close the performance gap to a state-of-the-art latent variational method (L3DVAR), surpassing it by 14.86% in z500 WRMSE under 5% observation coverage.

While this work introduces a novel approach that makes generative DA surpass variational DA, the nature of latent optimization, which requires multiple model evaluations per assimilation step, presents a significant computational cost compared to single-step methods. Despite the limitation, LO-SDA marks a paradigm shift in atmospheric DA. It serves as evidence that generative models, when integrated with variational principles, can effectively replace traditional Gaussian assumptions and significantly improve performance. By combining the flexibility of deep generative models with the rigorous constraints required in operational DA, LO-SDA represents a key step toward next-generation data assimilation systems. Future work should focus on improving computational efficiency and expanding the framework's applicability to more diverse and realistic atmospheric conditions.

## 6 ETHICS AND REPRODUCIBILITY STATEMENT

The authors have read and adhered to the ICLR Code of Ethics. This work utilizes publicly available datasets. We foresee no significant ethical concerns arising from this research.

To ensure reproducibility, we provide detailed information about our experimental setup, hyperparameters, and implementation in the appendix. The source code for this project will be made publicly available upon publication.

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

## A  USE OF LLMS STATEMENT

A large language model was used to assist with grammar and language refinement in this manuscript.

## B  WRMSE

The latitude-weighted root mean square error (WRMSE) is a statistical metric widely used in geospatial analysis and atmospheric science. Given the estimate $\hat{x}_{h,w,c}$ and the truth $x_{h,w,c}$, the WRMSE is defined as,

$$\text{WRMSE}(c) = \sqrt{\frac{1}{H \cdot W} \sum_{h,w} H \frac{\cos(\alpha_{h,w})}{\sum_{h'=1}^{H} \cos(\alpha_{h',w})} (x_{h,w,c} - \hat{x}_{h,w,c})^2} \ . \tag{16}$$

Here $H$ and $W$ represent the number of grid points in the longitudinal and latitudinal directions, respectively, and $\alpha_{h,w}$ is the latitude of point $(h, w)$.

## C  THE VAE TRAINING AND RESULTS

**Model structure and training** We utilize a transformer-based variational autoencoder framework (VAEformer) to effectively reduce the dimensionality of atmospheric data, mapping high-dimensional fields to a compact latent representation Han et al. (2024). The architecture incorporates window-based attention mechanisms Liu et al. (2021) to efficiently model atmospheric circulation patterns. Following the "vit_large" design paradigm, our implementation features identical encoder and decoder structures employing 4×4 patch embeddings with matching stride, a 1024-dimensional latent space, and a 24-layer transformer network utilizing window attention. The model was trained on ERA5 reanalysis data spanning 1979-2016, with the subsequent two-year period (2016-2018) serving as validation, over the course of 60 training epochs.

**Results** Our trained VAE achieves 0.0067 overall MSE and 0.0486 overall MAE. The varibles WRMSE are presented in Table 4.

Table 4: The VAE training results on WRMSE

| u10 | v10 | t2m | msl | z50 | z100 | z150 | z200 | z250 | z300 |
|---|---|---|---|---|---|---|---|---|---|
| 0.54832 | 0.50501 | 0.82944 | 34.002 | 75.529 | 55.645 | 42.436 | 38.426 | 35.963 | 35.008 |
| z400 | z500 | z600 | z700 | z850 | z925 | z1000 | q50 | q100 | q150 |
| 31.623 | 28.4 | 25.948 | 24.563 | 23.415 | 24.37 | 27.266 | 9.64E-09 | 6.35E-08 | 4.90E-07 |
| q200 | q250 | q300 | q400 | q500 | q600 | q700 | q850 | q925 | q1000 |
| 3.04E-06 | 1.02E-05 | 2.47E-05 | 7.81E-05 | 1.68E-04 | 2.73E-04 | 4.01E-04 | 6.02E-04 | 5.95E-04 | 4.69E-04 |
| u50 | u100 | u150 | u200 | u250 | u300 | u400 | u500 | u600 | u700 |
| 0.91052 | 1.1085 | 1.3769 | 1.5108 | 1.5418 | 1.5148 | 1.3712 | 1.2184 | 1.1193 | 1.0552 |
| u850 | u925 | u1000 | v50 | v100 | v150 | v200 | v250 | v300 | v400 |
| 0.95107 | 0.78308 | 0.60413 | 0.80967 | 0.91698 | 1.1081 | 1.2589 | 1.3588 | 1.3544 | 1.2148 |
| v500 | v600 | v700 | v850 | v925 | v1000 | t50 | t100 | t150 | t200 |
| 1.0672 | 0.96791 | 0.90445 | 0.84409 | 0.71597 | 0.55351 | 0.59292 | 0.64698 | 0.47627 | 0.39086 |
| t250 | t300 | t400 | t500 | t600 | t700 | t850 | t925 | t1000 | |
| 0.39115 | 0.41718 | 0.47806 | 0.48918 | 0.50497 | 0.5381 | 0.64627 | 0.61494 | 0.66865 | |

## D  RESAMPLING

Here we provide a derivation of Equation 13. Assume we have two independent Gaussian distributions:

$$p_a(x) = \mathcal{N}(x; \mu_a, \sigma_a^2) \tag{17}$$

$$p_b(x) = \mathcal{N}(x; \mu_b, \sigma_b^2) \tag{18}$$

The product distribution $p_c(x) = p_a(x)p_b(x)$ is also Gaussian with parameters:

$$\mu_c = \frac{\mu_a/\sigma_a^2 + \mu_b/\sigma_b^2}{1/\sigma_a^2 + 1/\sigma_b^2} \tag{19}$$

$$\sigma_c^2 = \frac{1}{1/\sigma_a^2 + 1/\sigma_b^2} \tag{20}$$

*Proof:*

$$p_c(x) \propto \exp\left(-\frac{(x-\mu_a)^2}{2\sigma_a^2}\right)\exp\left(-\frac{(x-\mu_b)^2}{2\sigma_b^2}\right)$$

$$= \exp\left(-\frac{1}{2}\left(\frac{1}{\sigma_a^2} + \frac{1}{\sigma_b^2}\right)x^2 + x\left(\frac{\mu_a}{\sigma_a^2} + \frac{\mu_b}{\sigma_b^2}\right) + C\right) \tag{21}$$

where $C$ contains terms independent of $x$. Completing the square, we obtain:

$$p_c(x) \propto \exp\left(-\frac{(x-\mu_c)^2}{2\sigma_c^2}\right) \tag{22}$$

with $\mu_c$ and $\sigma_c^2$ as defined above.

Now consider the conditional distribution in Equation 13, where:

$$p(\boldsymbol{z}_t|\hat{\boldsymbol{z}}_0(\boldsymbol{y}), \boldsymbol{y}) \sim \mathcal{N}\left(\mu(t)\hat{\boldsymbol{z}}_0(\boldsymbol{y}), \sigma^2(t)\boldsymbol{I}\right) \tag{23}$$

$$p(\tilde{\boldsymbol{z}}_t|\boldsymbol{z}_t, \hat{\boldsymbol{z}}_0(\boldsymbol{y}), \boldsymbol{y}) \sim \mathcal{N}(\boldsymbol{z}_t, \lambda_t^2\boldsymbol{I}) \tag{24}$$

The posterior distribution is given by:

$$p(\boldsymbol{z}_t = \boldsymbol{a}|\tilde{\boldsymbol{z}}_t, \hat{\boldsymbol{z}}_0(\boldsymbol{y}), \boldsymbol{y}) \propto p(\tilde{\boldsymbol{z}}_t|\boldsymbol{z}_t = \boldsymbol{a})p(\boldsymbol{z}_t = \boldsymbol{a}|\hat{\boldsymbol{z}}_0(\boldsymbol{y}), \boldsymbol{y})$$

$$\propto \exp\left(-\frac{\|\boldsymbol{a} - \tilde{\boldsymbol{z}}_t\|^2}{2\lambda_t^2}\right)\exp\left(-\frac{\|\boldsymbol{a} - \mu(t)\hat{\boldsymbol{z}}_0(\boldsymbol{y})\|^2}{2\sigma^2(t)}\right)$$

$$\propto \exp\left(-\frac{1}{2}\left(\frac{1}{\lambda_t^2} + \frac{1}{\sigma^2(t)}\right)\|\boldsymbol{a}\|^2 + \left\langle \boldsymbol{a}, \frac{\tilde{\boldsymbol{z}}_t}{\lambda_t^2} + \frac{\mu(t)\hat{\boldsymbol{z}}_0(\boldsymbol{y})}{\sigma^2(t)}\right\rangle\right) \tag{25}$$

Applying the product formula for Gaussians, we obtain:

$$p(\boldsymbol{z}_t|\tilde{\boldsymbol{z}}_t, \hat{\boldsymbol{z}}_0(\boldsymbol{y}), \boldsymbol{y}) \sim \mathcal{N}\left(\frac{\lambda_t^2\mu(t)\hat{\boldsymbol{z}}_0(\boldsymbol{y}) + \sigma^2(t)\tilde{\boldsymbol{z}}_t}{\lambda_t^2 + \sigma^2(t)}, \frac{\lambda_t^2\sigma^2(t)}{\lambda_t^2 + \sigma^2(t)}\boldsymbol{I}\right) \tag{26}$$

# E   REAL-WORLD OBSERVATIONS

To evaluate our framework under real-world conditions, we employ the Global Data Assimilation System (GDAS) prepbufr dataset, which incorporates multi-source observations. For this study, only surface and radiosonde observations are utilized. These observations are first interpolated onto the model state grid, and any multiple observations at a single grid point are averaged. High-elevation surface observations are vertically interpolated and reclassified as upper-air data. A quality control procedure is further applied to remove observations with large deviations. Observations are dropped if their deviation from the ERA5 reference exceeds 0.05 of the ERA5 climatological standard deviation. We perform data assimilation daily at 00:00 UTC throughout 2017, using a 48-hour background field. As shown in Table 5, the results indicate that LO-SDA achieves performance comparable to L3DVAR, and slightly outperforms the traditional 3DVAR when using real observations.

# F   STATISTICAL SIGNIFICANCE

To rigorously evaluate the statistical significance of our results, we present the mean errors and standard deviations across the whole test dataset in Table 6. This analysis confirms our main conclusions: the performance of LO-SDA is statistically comparable to the state-of-the-art L3DVAR method, particularly under 1% observation. Conversely, LO-SDA demonstrates a clear and statistically significant advantage over both traditional 3DVAR and other diffusion-based approaches (DPS and Repaint), as evidenced by the consistently lower mean errors and non-overlapping standard deviations. Notably, LO-SDA also exhibits greater stability, with smaller error variances than other generative methods, suggesting more robust performance across diverse atmospheric conditions.

Table 5: Assimilation performance on real-world observation across various methods.

| | MSE | MAE | WRMSE | | | | | |
|---|---|---|---|---|---|---|---|---|
| | | | msl | u10 | u700 | v500 | z500 | t850 |
| 48h background | 0.0475 | 0.1158 | 98.7910 | 1.2588 | 1.9692 | 2.4061 | 89.2800 | 0.9232 |
| 3DVAR | 0.0472 | 0.1150 | 87.1536 | 1.2532 | 1.9646 | 2.3950 | 83.8241 | 0.9068 |
| L3DVAR | 0.0467 | 0.1143 | 84.8751 | 1.2376 | 1.9597 | 2.3778 | 78.1842 | 0.8962 |
| LO-SDA(ours) | 0.0469 | 0.1140 | 81.4013 | 1.2128 | 1.9891 | 2.3657 | 77.1881 | 0.9914 |

Table 6: The statistical significance analysis of different methods under 1% and 5% observations. The red and yellow indicate the first and second best performing methods.

| Ratio | Model | WRMSE | | | | | |
|---|---|---|---|---|---|---|---|
| | | msl | u10 | u700 | v500 | z500 | t850 |
| 1% observation | 3DVAR | $81.6384 \pm 5.1771$ | $1.2235 \pm 0.0509$ | $1.9850 \pm 0.0755$ | $2.4298 \pm 0.1022$ | $62.9377 \pm 4.5379$ | $0.8797 \pm 0.0304$ |
| | L3DVAR | $62.1054 \pm 3.2258$ | $1.1862 \pm 0.0435$ | $1.9797 \pm 0.0682$ | $2.3392 \pm 0.0928$ | $53.0902 \pm 2.8705$ | $0.8975 \pm 0.0289$ |
| | Repaint | $114.3672 \pm 6.3959$ | $1.4167 \pm 0.0525$ | $2.1059 \pm 0.0716$ | $2.5664 \pm 0.1081$ | $104.1351 \pm 7.2029$ | $1.0363 \pm 0.03345$ |
| | DPS | $95.9850 \pm 5.4456$ | $1.3286 \pm 0.0453$ | $2.0220 \pm 0.0764$ | $2.3981 \pm 0.0887$ | $85.5673 \pm 11.1945$ | $1.0269 \pm 0.0412$ |
| | LO-SDA(ours) | $62.4505 \pm 3.0864$ | $1.1836 \pm 0.0387$ | $1.8981 \pm 0.0597$ | $2.2439 \pm 0.0836$ | $53.6468 \pm 2.6033$ | $0.9243 \pm 0.0274$ |
| 5% observation | 3DVAR | $63.2562 \pm 3.5630$ | $1.1661 \pm 0.0484$ | $1.8765 \pm 0.0864$ | $2.2365 \pm 0.1173$ | $45.8574 \pm 2.5304$ | $0.7849 \pm 0.0259$ |
| | L3DVAR | $45.8037 \pm 1.6854$ | $0.9350 \pm 0.0308$ | $1.7166 \pm 0.0533$ | $1.9400 \pm 0.0713$ | $38.6411 \pm 1.6761$ | $0.8024 \pm 0.0242$ |
| | Repaint | $106.3938 \pm 5.9038$ | $1.2934 \pm 0.0455$ | $1.9889 \pm 0.0648$ | $2.3622 \pm 0.0975$ | $95.9167 \pm 6.7577$ | $0.9856 \pm 0.0315$ |
| | DPS | $93.2470 \pm 4.0291$ | $1.2673 \pm 0.0429$ | $1.9585 \pm 0.0657$ | $2.3271 \pm 0.0910$ | $85.3329 \pm 9.0282$ | $0.9891 \pm 0.0331$ |
| | LO-SDA(ours) | $42.3498 \pm 1.5015$ | $0.8992 \pm 0.2159$ | $1.5873 \pm 0.0445$ | $1.7894 \pm 0.0615$ | $32.8990 \pm 1.3194$ | $0.8094 \pm 0.2179$ |

# G CONVERGENCE ANALYSIS AND COMPUTATIONAL EFFICIENCY

To address the practical concerns regarding computational cost, we conducted a comprehensive ablation study to analyze the convergence behavior of LO-SDA with respect to the number of diffusion sampling steps and latent optimization iterations. As illustrated in Figure 3, the assimilation error (MSE) converges rapidly, stabilizing after approximately 32 sampling steps and 50 optimization iterations. By adopting these optimized hyperparameters, according to the runtime breakdown in Table 7, we can reduce the inference runtime to approximately 1 minute per analysis with negligible performance loss. This confirms that LO-SDA is computationally efficient and scalable for operational settings.

Table 7: Runtime breakdown of LO-SDA for a single analysis cycle. The configuration uses 64 diffusion steps, with latent optimization applied at 25% of the steps (16 steps) for 50 iterations each.

| Component | Configuration | Time Cost | Ratio |
|---|---|---|---|
| Diffusion Sampling | 48 steps $\times 0.15$s | 7.2 s | 8.3% |
| Latent Optimization | 16 steps $\times$ 50 iters $\times$ 0.10s | 80.0 s | 91.7% |
| VAE Encode/Decode | Included in above | Negligible | less than 1% |
| **Total Runtime** | - | **87.2 s** | **100%** |

# H CYCLIC DA

To evaluate the LO-SDA's stability for operational forecasting cycles, we conducted a year-long (2019) cyclical data assimilation experiment (assimilation every 48 hours) with 5% observation density. As shown in Figure ??, the analysis RMSE remains consistently low and stable throughout the entire year, exhibiting no signs of error accumulation or filter divergence. This empirical evidence demonstrates that LO-SDA is not limited to single-step analysis but is fully capable of handling continuous, long-term sequential data assimilation tasks with high stability.

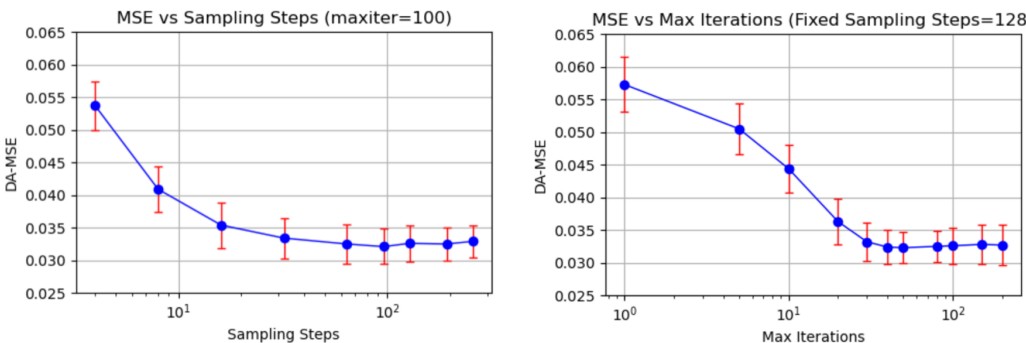

Figure 3: The empirical convergence analysis/

# I   NOISE ROBUSTNESS ANALYSIS

Our framework employs a hyperparameter $\tau$ in the latent optimization loop to control the strictness of the observation constraint. A smaller $\tau$ enforces tighter consistency with observations, while the larger one relaxes this constraint, placing more reliance on the diffusion prior. As shown in Table 8, under high-noise conditions, the default strict constraint ($\tau = 0.0001$) leads to slight overfitting of the noise, resulting in performance comparable to 3DVAR (shown in main text). However, by simply relaxing the constraint threshold to $\tau = 0.001$, LO-SDA achieves a balance between noise filtering and data assimilation. The tuned model yields the lowest DA error, outperforming both the traditional 3DVAR and the competitive L3DVAR baseline. This demonstrates that LO-SDA possesses inherent flexibility to handle varying data quality by adjusting the optimization strength.

Table 8: Quantitative performance comparison under high-noise conditions (std = 0.10). The LO-SDA framework demonstrates robustness by adjusting the optimization constraint parameter $\tau$. Best results are highlighted in **bold**.

| Ratio | Model | MSE | MAE | WRMSE | | | | | |
| --- | --- | --- | --- | --- | --- | --- | --- | --- | --- |
| | | | | msl | u10 | u700 | v500 | z500 | t850 |
| | 48h background | 0.0505 | 0.1178 | 98.7265 | 1.2727 | 1.9953 | 2.4217 | 89.2752 | 0.9310 |
| std = 0.10 | 3DVAR | 0.0495 | 0.1173 | 91.2331 | 1.2291 | 1.9493 | 2.3636 | 84.4765 | 0.9101 |
| | L3DVAR | 0.0489 | 0.1147 | 83.1325 | 1.1954 | 1.9448 | 2.3275 | 75.3260 | 0.9341 |
| | LO-SDA($\tau = 0.001$) | **0.0481** | **0.1130** | **79.1052** | **1.1831** | **1.9254** | **2.2804** | 69.8194 | **0.8965** |
| | LO-SDA($\tau = 0.0001$) | 0.0498 | 0.1188 | 82.0772 | 1.2408 | 1.9602 | 2.3194 | **69.4214** | 1.0308 |

# J   MORE VISUALIZATION.

We provide additional visualization results comparing different assimilation methods under 5% observation. In all appendix figures, the top row (left to right) displays the ERA5 ground truth, background field, and background error. The middle row shows assimilation results from (a) our proposed LO-SDA method, (b) the DPS framework, and (c) the Repaint approach, while the bottom row presents the corresponding absolute error fields relative to ERA5 truth. The significantly lighter error magnitudes in the LO-SDA results highlight our method's superior error reduction capability compared to alternative approaches.

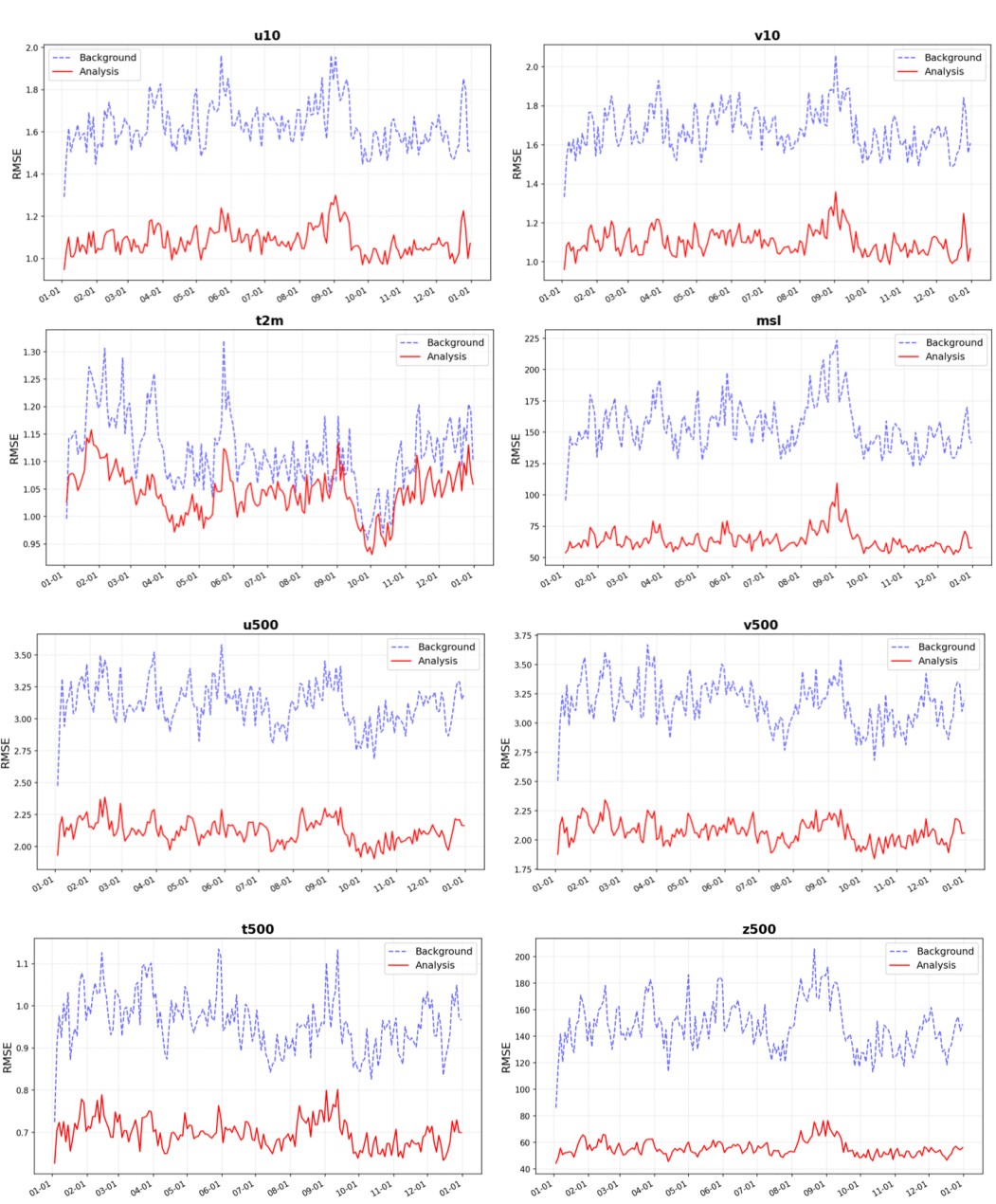

Figure 4: The cyclic DA across the entire 2019.

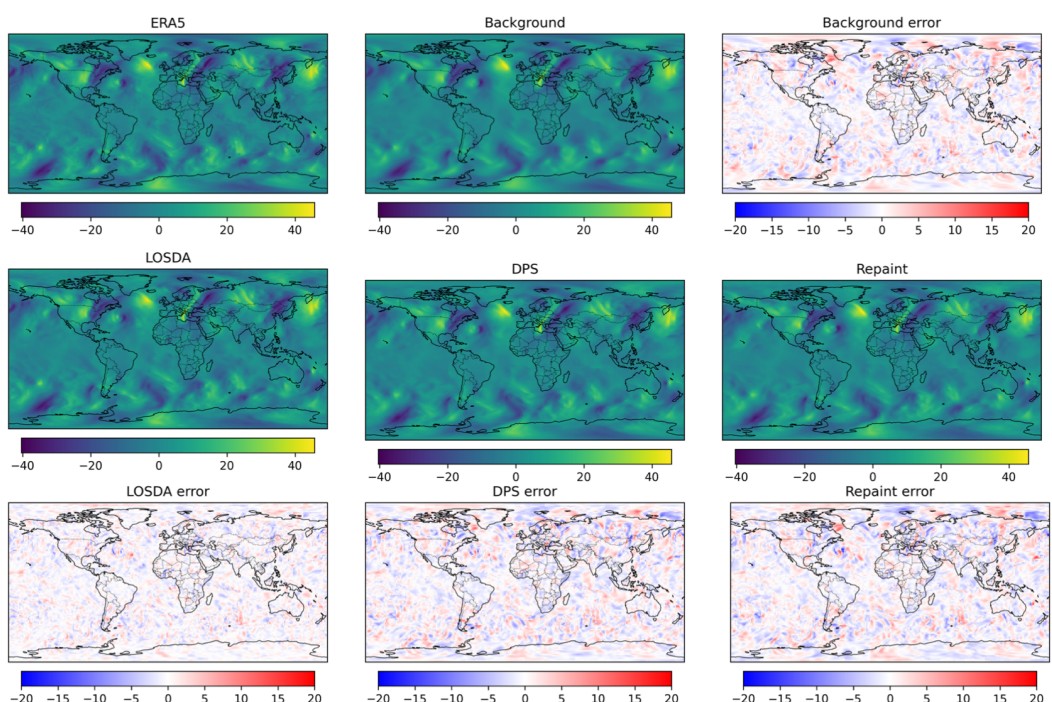

Figure 5: Visulaization of u500 at a 2019-08-26-06:00 UTC.

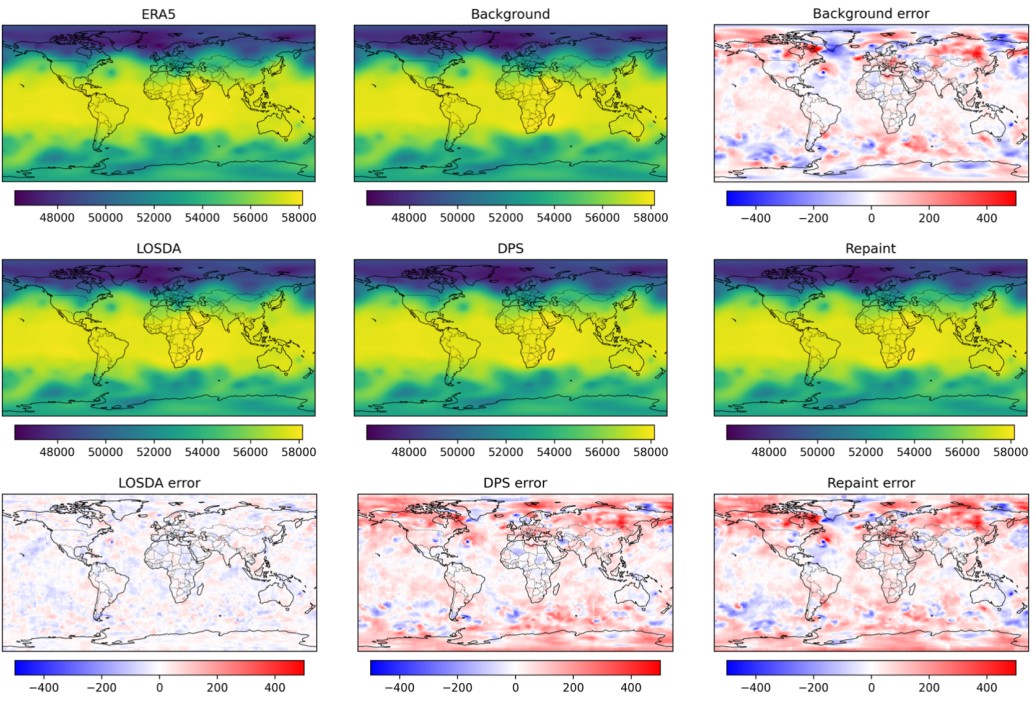

Figure 6: Visulaization of z500 at a 2019-05-18-06:00 UTC.

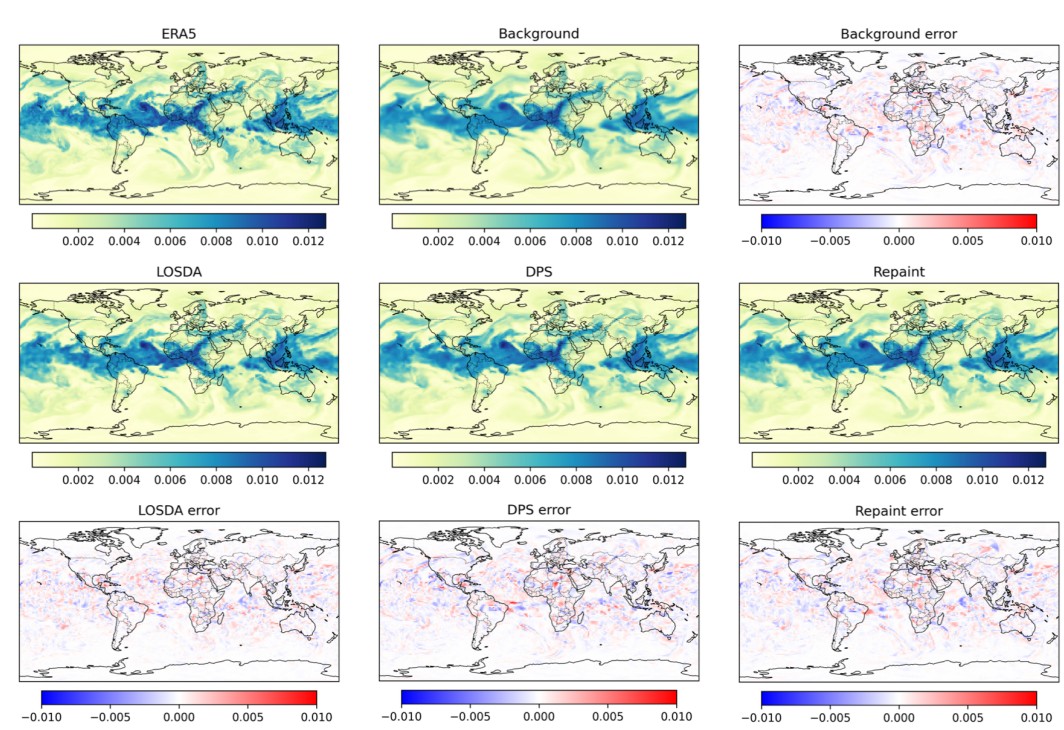

Figure 7: Visulaization of q700 at a 2019-02-02-06:00 UTC.

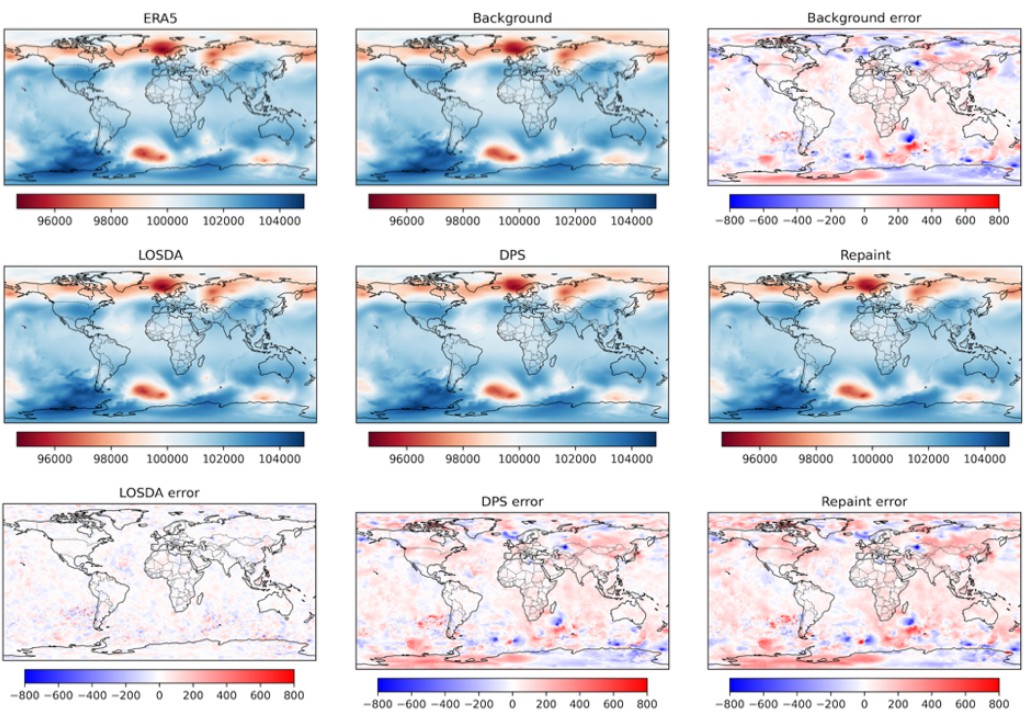

Figure 8: Visulaization of msl at a 2019-04-07-06:00 UTC.

