# OpenReview forum: "LO-SDA: Latent Optimization for Score-based Atmospheric Data Assimilation"
_ICLR.cc/2026/Conference — Submitted to ICLR 2026_

### Official Review · Reviewer_ZLd6 · 2025-10-14

**Soundness:** 2
**Presentation:** 2
**Contribution:** 1
**Rating:** 2
**Confidence:** 4

**Summary:**

First of all, I would like to state that I have previously been randomly assigned to this paper for a prior conference as well. If there are some conflicts of interests here, I humbly request the area chairs to take this into account and/or remove the review if needed. I may reference some of my previous points if they were left unaddressed in this updated version of the manuscript.


The authors create a background-conditioned latent diffusion prior inside a VAE, such that they minimize using a variational method to optimize for the observations. During the reverse pass, they apply alternating latent optimisation steps that explicitly minimise the observation-error cost.

**Strengths:**

The model improves over other methods such as Repaint, DPS, and 3DVar-esque techniques. Seems to have a respectable improvement under 5% observations, but for 1% observations, the improvements over other methods seem marginal at best (given the error bars). Thank you for updating the manuscript with some uncertainty/error bars so we can more closely look at the trends!

They test against both 1% and 5% sparse observations conditions, demonstrating that the model works for both. However, the trend seems as though it might not scale well with higher sparsity scenarios (<1%) relative to the other methods.

Due to optimizing in a latent space, the model should be more computationally efficient at assimilation for high-dimensional data. However, though this claim is made, there is no comparison of runtime with respect to traditional DA methods.

**Weaknesses:**

The failure mode of repaint (compared to the background) is still not explored in this revised edition of the paper. Perhaps a small discussion would help strengthen the results by establishing the baseline in a more proper form.

As mentioned before, there is some limited novelty of the paper compared to recent diffusion-based data assimilation approaches such as APPA https://arxiv.org/abs/2504.18720, latent diffusion DA https://arxiv.org/abs/2406.14815, etc. Perhaps some discussion on the particular impacts of this work relative to some of the aforementioned papers would strengthen this paper.

The model seems to have a lot more variability due to noise uncertainty, as portrayed in Table 2, showing that it might not extrapolate to higher noise regimes. Is there a specific explanation as for such?

Also, I would still recommend adding a small note for the red and yellow highlights for first/second best.

**Questions:**

I'm not actually sure whether the DA is performed repeatedly or only once after 48h after calculating the background? If so, how does the background distribution changing due to the data assimilation process itself actually affect the DA procedure later down the line? If not, then this would additionally need to be tested.

What differentiates this model from previous score-based data assimilation techniques (papers listed above)? Additionally, have the authors have tried any of the previous latent diffusion data assimilation models as baselines?

---

> ### Author Response · Authors · 2025-11-27
>
> We sincerely thank the reviewer for the continued engagement. We have made substantial revisions to address your concerns. We hope that this revision will be satisfactory.
>
>
> ___
>
> > **Response for Question** **1:** Is the DA performed repeatedly or only once?
>
>   We would like to clarify that our experiments include only single-step assimilation, not cyclic assimilation. The primary goal of this work was to demonstrate the advantages of using diffusion models to represent the prior compared with traditional DA algorithms. Therefore, using the same background fields in the experiments ensures a fair and consistent comparison across methods.
>
> Nevertheless, inspired by your suggestion, we conducted a new, extensive **year-long cyclical DA experiment** with 5% observation denstiy and every 48h assimilation for all of 2019. The results, now added to the Appendix, are extremely promising. As shown in the new figure and below table, the analysis RMSE for all key variables **remains stable and consistently low throughout the entire year, exhibiting no signs of significant error accumulation or** **divergence**.
>
> | Var | Mean RMSE (Bg -> **Ours**) | Std Dev (Bg -> **Ours**) | Max Peak (Bg -> **Ours**) | Imp. |
> | :--- | :---: | :---: | :---: | :---: |
> | **z500** | 147.7 $\to$ **55.6** | 18.5 $\to$ **5.7** | 205.7 $\to$ **76.3** | **62.4%** |
> | **u500** | 3.12 $\to$ **2.12** | 0.17 $\to$ **0.09** | 3.58 $\to$ **2.39** | **32.0%** |
> | **t500** | 0.96 $\to$ **0.70** | 0.07 $\to$ **0.03** | 1.13 $\to$ **0.80** | **27.3%** |
> | **v500** | 3.17 $\to$ **2.07** | 0.19 $\to$ **0.10** | 3.67 $\to$ **2.34** | **34.7%** |
> | **u10** | 1.63 $\to$ **1.08** | 0.11 $\to$ **0.06** | 1.96 $\to$ **1.30** | **33.5%** |
> | **v10** | 1.68 $\to$ **1.10** | 0.11 $\to$ **0.06** | 2.06 $\to$ **1.36** | **34.2%** |
> | **t2m** | 1.12 $\to$ **1.04** | 0.07 $\to$ **0.04** | 1.32 $\to$ **1.16** | **6.5%** |
> | **msl** | 154.96 $\to$ **63.85** | 19.70 $\to$ **8.28** | 223.28 $\to$ **109.32** | **58.8%** |
>
> This provides strong empirical evidence for the long-term stability and effectiveness of LO-SDA in a more realistic, sequential setting. We are grateful for your question, as this further strengthens our paper's contribution by demonstrating the long-term robustness of our method.
>
>
> ___
>
> > **Response for Question** **2 and Weakness 2:** Comparison to other SDA and Baselines
>
> Thank you for mentioning other SDA works. Our approach has two critical distinctions from methods like APPA.
>
> The first point is how we model the prior. APPA learns an **unconditional prior** of atmospheric trajectories, acting as a combined forecast and assimilation system. In contrast, LO-SDA learns a **background-conditioned prior**, designed to model the non-Gaussian error of a background field from an external forecast model. Both traditional DA and AI-based atmospheric DA have shown that incorporating a forecast as the background field is crucial []. In addition, this approach can, in principle, be coupled with traditional numerical weather prediction systems and replace traditional DA module, whereas APPA cannot.
>
>  The second and most important contribution is our method for integrating observations. Previous methods like APPA and DiffDA use **"single-step gradient guidance"** (We label it as DPS in our paper), which we found may not compete with traditional data assimilation algorithms due to insufficient observational constraints. Our work moves to **"iterative latent optimization,"** which explicitly minimizes an observation cost function at multiple steps. This process is a generative analogue to the rigorous cost minimization in variational DA, enforcing constraints much more strictly. This key mechanism enabled LO-SDA to outperform the DPS baseline significantly and, for the first time, surpasses the traditional 3DVar data assimilation algorithm. We have emphasized in the manuscript that DPS is the commonly used scheme in existing SDA studies, and we also highlight its limitation in providing insufficient observational constraints.

---

> ### Author Response · Authors · 2025-11-27
>
> ___
>
> > **Response for Weakness 1:** Failure mode of Repaint
>
> We thank the reviewer for this crucial point. The core limitation of Repaint in sparse-observation settings, as also demonstrated in DiffDA (Huang et al., 2024), lies in its inadequate enforcement of observational constraints. Two factors might cause this. First, in highly sparse scenarios, the weak observational signal is easily overpowered by the diffusion model's strong generative prior, failing to correct the state. Second, imposing a few sparse, disconnected observation points creates an "unnatural" input for the denoiser, which is trained on globally coherent fields. This mismatch can generate artifacts in unobserved regions, introducing errors that outweigh the benefit of fitting the few observations, sometimes leading to a higher RMSE than the background fields. This analysis highlights the necessity for a more robust mechanism like our iterative optimization, and we will add this detailed discussion to Section 4.1.
>
> ___
>
> > **Response for Weakness 3:** More variability due to noise uncertainty
>
> We appreciate this valuable observation. Our experiments in Table 2 already demonstrate LO-SDA's strong robustness to observation noise.  The variational methods (3DVar and L3DVar) account for the observation noise explicitly through the observation error covariance matrix. In contrast, although our LO-SDA is designed to enforce observation consistency,  we introduce a hyperparameter \tau in the optimization loop, which controls the early-stopping threshold and thus the "strength" of the observation enforcement. In our paper, we fixed \tau=0.0001 without exhausting tuning.  Inspired by your suggestion, we conducted a new experiment for the high-noise (`std=0.10`) scenario by relaxing this constraint (setting `τ=1e-3` instead of the original `1e-4`).
>
> | Ratio      | Model                 | MSE    | MAE    | WRMSE msl | WRMSE u10 | WRMSE u700 | WRMSE v500 | WRMSE z500 | WRMSE t850 |
> | ---------- | --------------------- | ------ | ------ | --------- | --------- | ---------- | ---------- | ---------- | ---------- |
> |            | 48h background        | 0.0505 | 0.1178 | 98.7265   | 1.2727    | 1.9953     | 2.4217     | 89.2752    | 0.931      |
> | std = 0.10 | 3DVAR                 | 0.0495 | 0.1173 | 91.2331   | 1.2291    | 1.9493     | 2.3636     | 84.4765    | 0.9101     |
> |            | L3DVAR                | 0.0489 | 0.1147 | 83.1325   | 1.1954    | 1.9448     | 2.3275     | 75.326     | 0.9341     |
> |            | LO-SDA($\tau=0.0010$) | 0.0481 | 0.113  | 79.1052   | 1.1831    | 1.9254     | 2.2804     | 69.8194    | 0.8965     |
> |            | LO-SDA($\tau=0.0001$) | 0.0498 | 0.1188 | 82.0772   | 1.2408    | 1.9602     | 2.3194     | 69.4214    | 1.0308     |
>
> The above table demonstrates that the observed variability is a controllable aspect of our framework. By tuning `τ`, LO-SDA can flexibly balance data consistency with noise robustness. We are very grateful for your question, as it has allowed us to uncover and highlight this important, adaptable feature of our method, significantly strengthening the paper.
>
> ___
>
> > **Response for Weakness 4:** Note for the red and yellow highlights
>
> Thank you; we have added a note to all tables clarifying that red and yellow indicate the first and second best performing methods.
>
>
>
> ___
>
> We sincerely hope that the clarifications provided above have addressed the concerns and questions raised. We remain open to further scrutiny and are eager to receive your feedback on whether these responses adequately address the points of consideration.

---

### Official Review · Reviewer_kg13 · 2025-10-29

**Soundness:** 2
**Presentation:** 2
**Contribution:** 2
**Rating:** 4
**Confidence:** 4

**Summary:**

This paper introduces LO-SDA, a novel generative data assimilation method designed for numerical weather prediction. The key idea is to relax the Gaussian prior assumptions of traditional data assimilation by leveraging score-based generative modeling in a learned latent space. First, a variational autoencoder (VAE) is trained to encode high-dimensional atmospheric states into a compact latent representation. Then a diffusion model (score-based generator) conditioned on the background state is used to sample from an approximate posterior in this latent space. During the reverse diffusion process, the method applies latent optimization to enforce consistency with sparse observations, effectively nudging the generated latent state to agree with observed values.

The contributions include: (1) a new latent-space diffusion assimilation framework that can handle highly sparse observations by combining VAE compression, conditional diffusion sampling, and a novel observation-constrained optimization step; and (2) an experimental demonstration on an idealized global atmospheric dataset showing that this generative approach can match or surpass the accuracy of state-of-the-art variational methods. Notably, under extremely sparse observation coverage (only 1% of state variables observed), LO-SDA achieves comparable performance to a leading latent variational baseline (L3DVar). As observation density increases (e.g. 5% coverage), LO-SDA decisively outperforms the baseline, achieving about a 15% reduction in error (WRMSE). This is presented as the first instance where a diffusion-based data assimilation method demonstrates the potential to outperform traditional Gaussian-based methods at a global scale. The results suggest that generative modeling techniques, when properly constrained by observations, can overcome some limitations of classical methods and improve assimilation accuracy in challenging, high-dimensional settings.

**Strengths:**

Originality: The paper proposes a creative combination of deep generative modeling with data assimilation. Using a conditional diffusion model in a VAE latent space to perform Bayesian inference is a novel approach in the context of atmospheric data assimilation. The introduction of a latent optimization step to enforce observation consistency is an innovative twist that addresses a known issue in score-based filters (i.e., ensuring the analysis state actually fits the sparse observations).

Quality: The methodology builds on recent advances in both data assimilation and generative modeling. The authors validate the approach on a high-dimensional global atmosphere simulation, which lends credibility that the method can scale to realistic scenarios. The experiments compare LO-SDA against appropriate baselines: a traditional 3D-Var and a learned latent 3D-Var (L3DVar) method, as well as other score-based sampling approaches (e.g., diffusion posterior sampling and the Repaint technique). The results show LO-SDA consistently performs well, particularly as observation density increases, and it outperforms both the traditional variational method and prior score-based methods in most settings. The paper also provides statistical significance analysis to support the improvements, adding rigor to the empirical findings.

Clarity: The paper is generally well-written and structured. It motivates the problem (limitations of Gaussian priors in DA) clearly and provides sufficient background on data assimilation and score-based models for readability. The algorithmic description of LO-SDA is detailed, and the roles of the VAE and diffusion model are explained with appropriate equations. Figures (if any, e.g. illustrating the model architecture or results) are clear. Overall, the authors do a good job making a complex method accessible to readers from both the machine learning and data assimilation communities.

Significance: If the claims hold, this work is quite significant for data assimilation research. It challenges the long-standing reliance on Gaussian assumptions by showing that a learned generative prior can surpass a state-of-the-art variational DA method on a large-scale problem. This suggests a potential paradigm shift in how operational weather forecasting systems could incorporate machine learning. Moreover, LO-SDA’s strong performance under extremely sparse observations is important—sparse observational data is a common scenario (e.g., in parts of the globe or for certain variables), and methods that can handle such settings are highly valuable. By demonstrating an approach that remains accurate with as low as 1% observation coverage, the paper addresses a critical challenge in high-dimensional Bayesian filtering. The techniques introduced here (latent-space assimilation with diffusion models) could inspire further research and development of new hybrid DA algorithms in both the geosciences and broader sequential data domains.

**Weaknesses:**

Baseline Comparisons: A notable weakness is the absence of comparison with some established DA methods like the Ensemble Kalman Filter (e.g., LETKF). LETKF and similar ensemble Kalman techniques are widely used operationally and are strong benchmarks for assimilation performance. The paper claims to outperform “traditional approaches,” but this is only demonstrated against 3DVar/L3DVar. Without experiments against an ensemble Kalman filter, it is unclear how LO-SDA stands relative to the true state-of-practice. This missing baseline makes the empirical evaluation feel incomplete and may overstate the improvements of LO-SDA.

Improvement Magnitude: While LO-SDA does outperform the latent 3DVar (L3DVar) at higher observation densities, the gains under very sparse observations are quite modest. In the toughest scenario (1% observations), LO-SDA’s error is essentially on par with L3DVar. This suggests that the new method, despite its sophisticated generative approach, does not dramatically outshine the variational baseline when data are extremely scarce. The paper should be careful not to overclaim superiority; the results indicate LO-SDA’s advantage emerges mainly when there are a bit more observations (5% or above), whereas in the most data-starved case its performance difference is negligible. This limits the practical advantage in scenarios of extreme sparsity, which is supposedly a key motivation.

Related Work and Novelty: The authors do not discuss or cite several closely related approaches in latent-space and generative data assimilation. For example, Latent-EnSF
arxiv.org
 and LD-EnSF
arxiv.org
 are recent methods that also perform assimilation in a learned latent space to handle high-dimensional states with sparse observations (using ensemble score-based filtering). These works similarly leverage VAEs to encode states and observations and have demonstrated improved accuracy and efficiency in scenarios with very sparse data. The submission’s technical approach (using a VAE for state compression and doing assimilation in latent space) is quite aligned with those, though using a diffusion model instead of an ensemble score matching. Failing to compare with or even acknowledge Latent-EnSF(https://arxiv.org/abs/2409.00127) / LD-EnSF( https://arxiv.org/abs/2411.19305 ) is a significant oversight. It raises questions about the novelty: the idea of latent-space data assimilation for sparse observations is not entirely new, and the paper should clarify how LO-SDA advances the state of the art relative to these methods. Similarly, there has been work on Deep Bayesian Filters (DBF https://arxiv.org/abs/2405.18674) for data assimilation that introduce learned latent dynamics and aim to perform Bayes-optimal sequential inference (e.g., by constructing latent variables and using neural networks for filtering). The authors do not compare to such approaches either, leaving a gap in understanding how LO-SDA compares to other modern ML-based filtering methods beyond the variational ones.

Theoretical Limitations: From a Bayesian perspective, LO-SDA has some theoretical limitations that the paper does not fully address. Notably, the method does not sample the true posterior distribution of the state given observations; it uses a diffusion model trained on data and then applies a deterministic optimization to fit observations. This procedure may break the theoretical guarantees of Bayesian inference – the result is an analysis that is a single point estimate consistent with observations, but not necessarily a representative sample from the posterior uncertainty. As a consequence, the approach might underestimate uncertainty or fail to explore multiple possible solutions when the observations are sparse/ambiguous. Additionally, the method’s design raises concerns about temporal consistency in sequential assimilation: each assimilation cycle involves running the diffusion backward with an optimization, which does not obviously ensure consistency over time steps. There is no explicit mechanism to enforce that the analysis at one time is statistically consistent with the background distribution at the next time. In contrast, standard Bayes filtering or sequential Monte Carlo methods propagate uncertainty forward; here it seems each cycle is solved somewhat independently in latent space. This could potentially lead to filter inconsistency or drift over long sequences. The paper would benefit from a discussion on these theoretical aspects – for example, what are the implications of using this one-shot latent optimization per cycle, and could it be integrated into a more principled sequential Bayesian framework?

Scope of Experiments: The experiments, while promising, are conducted on an idealized global atmospheric model setup. It’s not entirely clear how complex or realistic the model is (e.g., resolution, model physics) and whether the performance would hold in a fully operational NWP setting. Also, the computational cost of LO-SDA is not thoroughly discussed; diffusion models can be expensive, and although latent space reduces dimension, one wonders if the method is efficient enough for real-time forecasting. Some clarification on runtime or complexity compared to baselines would strengthen the evaluation. These are minor issues, but addressing them would give a more complete picture of the method’s practical viability.

Pretraining Assumptions and Generalization: A further limitation lies in the pretraining assumptions behind the VAE encoder. LO-SDA requires a VAE trained on fully observed high-dimensional atmospheric states to learn the latent space. However, in many real-world applications, such full-state ground truth is unavailable or prohibitively expensive to collect. It is unclear how well LO-SDA would generalize to settings where only partial or noisy state estimates are available during training, which is often the case in operational meteorology. Moreover, the method’s effectiveness hinges on the quality of the learned latent representation. If the VAE is trained on idealized or unrealistic data, the resulting latent space may not be suitable for use in sparse, noisy, or changing environments. Addressing how LO-SDA can be adapted to such partially observed training regimes, or evaluating its robustness to domain shifts in latent modeling, would strengthen its practical relevance.

**Questions:**

Related Work: How does LO-SDA compare to existing latent-space assimilation methods like Latent-EnSF and LD-EnSF? These approaches also target high-dimensional systems with sparse observations using VAEs and score-based updates. It would be helpful if the authors could explain the differences and why LO-SDA is expected to perform better. Was there a reason these methods were omitted from discussion? A clear positioning relative to them would strengthen the paper’s originality claim.

Sequential Filtering: The proposed method handles one assimilation step via diffusion + optimization, but how is the temporal sequence of assimilation cycles handled? Is LO-SDA applied independently at each cycle with a newly encoded background, or is there any mechanism to propagate information forward (e.g., using the posterior as the next background)? Essentially, is LO-SDA intended as a one-step analysis technique or part of a full filtering framework? If the latter, do the authors observe any issues with filter consistency or error accumulation over long sequences?

Relatedly, how does the method perform when a single observation time is insufficient to constrain the latent state, such as in scenarios with high observation noise or highly underdetermined inverse problems? Are there any experiments that test LO-SDA's robustness in these noisy or ill-posed conditions, where temporal information or smoothing would typically be necessary to resolve ambiguity?

Performance at Extreme Sparsity: The results show only a slight edge (or parity) over L3DVar at 1% observation coverage. Do the authors have insight into why the improvement is small in this regime? Is the diffusion model struggling due to very weak observational constraints, or could it be that L3DVar is already very effective at low obs densities? Understanding this would help assess LO-SDA’s usefulness in the scarcest-data scenarios. Also, if possible, what modifications might improve performance when observations are extremely sparse? (e.g., more latent optimization steps, better priors, etc.)

Computational Cost: What is the runtime of LO-SDA for a single analysis, and how does it compare to L3DVar or other baselines? The paper mentions latent 3DVar takes on the order of seconds per assimilation. If LO-SDA is significantly slower due to diffusion sampling, it might be a concern for operational use. Any details on optimization steps or potential to speed it up (like using fewer diffusion steps or a more efficient sampler) would be helpful to know.

---

> ### Author Response · Authors · 2025-11-27
>
> We sincerely thank the reviewer for their thorough, insightful, and highly constructive review. We are especially grateful for your positive assessment of our work's **Originality, Quality, Clarity, and Significance**. Your detailed feedback is invaluable, and we have made substantial revisions to the manuscript to address the weaknesses and questions you raised.
> ___
> > **Response for Question 1 & Weakness 3:** Related Work and Novelty (Comparison to Latent-EnSF, LD-EnSF, DBF)
>
> Thanks for mentioning these important relevant works. Our work shares the goal of latent-space data assimilation with recent methods, but is distinguished by its unique hybrid approach. Unlike the **ensemble-based** Latent-EnSF and LD-EnSF, which use standard score guidance, LO-SDA leverages a **single, more expressive diffusion model** as the prior and introduces **iterative optimization** to rigorously enforce observation consistency, inspired by variational principles. In contrast to Deep Bayesian Filter (DBF), which relies on **variational inference** and enforces **linear latent dynamics** for analytical tractability, our score-based framework makes no such structural assumptions. LO-SDA thus occupies a unique position, combining a powerful, non-Gaussian generative prior with the strict, optimization-based constraint enforcement of variational methods. We will update the manuscript to discuss these distinctions, positioning LO-SDA's novelty in its unique, variational-inspired integration of optimization into a generative non-Gaussian prior.
> ___
> > **Response for Question 2:** Sequential Filtering and Temporal Consistency
>
> We understand your question about long-term stability, which is a critical point. To address it directly, we conducted a new, **year-long cyclical data assimilation experiment**. In the following table, the significantly reduced RMSEs demonstrate the effectiveness of LO-SDA, while the drastically low standard deviations and suppressed peak errors confirm its exceptional stability in long-term cyclic assimilation. This provides strong empirical evidence that LO-SDA is robust and consistent for sequential applications, directly addressing your concern.
>
> | Var | Mean RMSE (Bg -> **Ours**) | Std Dev (Bg -> **Ours**) | Max Peak (Bg -> **Ours**) | Imp. |
> | :--- | :---: | :---: | :---: | :---: |
> | **z500** | 147.7 $\to$ **55.6** | 18.5 $\to$ **5.7** | 205.7 $\to$ **76.3** | **62.4%** |
> | **u500** | 3.12 $\to$ **2.12** | 0.17 $\to$ **0.09** | 3.58 $\to$ **2.39** | **32.0%** |
> | **t500** | 0.96 $\to$ **0.70** | 0.07 $\to$ **0.03** | 1.13 $\to$ **0.80** | **27.3%** |
> | **v500** | 3.17 $\to$ **2.07** | 0.19 $\to$ **0.10** | 3.67 $\to$ **2.34** | **34.7%** |
> | **u10** | 1.63 $\to$ **1.08** | 0.11 $\to$ **0.06** | 1.96 $\to$ **1.30** | **33.5%** |
> | **v10** | 1.68 $\to$ **1.10** | 0.11 $\to$ **0.06** | 2.06 $\to$ **1.36** | **34.2%** |
> | **t2m** | 1.12 $\to$ **1.04** | 0.07 $\to$ **0.04** | 1.32 $\to$ **1.16** | **6.5%** |
> | **msl** | 154.96 $\to$ **63.85** | 19.70 $\to$ **8.28** | 223.28 $\to$ **109.32** | **58.8%** |
> ___
> > **Response for Question 3:** Robustness to Noise
>
> We appreciate you raising this point. Our experiments in Table 2 already demonstrate LO-SDA's strong robustness across various noise levels. Furthermore, our framework offers inherent flexibility in handling noisy observations, which we believe is a significant advantage. While variational methods use the R matrix to model noise, LO-SDA can achieve a similar effect at inference time by adjusting the intensity of the latent optimization (e.g., the early stopping threshold, or a guidance scale). To validate this, we conducted a **new experiment in the highest noise setting (0.1 std)**. By slightly relaxing the optimization constraints, we achieved the balance between diffusion prior and observation enforcement, leading to **even better performance** than reported in the main text. This confirms that our method can be flexibly adapted to varying observational quality without retraining.
>
> | Ratio | Model | MSE | MAE | WRMSE msl | WRMSE u10 | WRMSE u700 | WRMSE v500 | WRMSE z500 | WRMSE t850 |
> | :--- | :--- | :--- | :--- | :--- | :--- | :--- | :--- | :--- | :--- |
> | | 48h background | 0.0505 | 0.1178 | 98.7265 | 1.2727 | 1.9953 | 2.4217 | 89.2752 | 0.931 |
> | std = 0.10 | 3DVAR | 0.0495 | 0.1173 | 91.2331 | 1.2291 | 1.9493 | 2.3636 | 84.4765 | 0.9101 |
> | | L3DVAR | 0.0489 | 0.1147 | 83.1325 | 1.1954 | 1.9448 | 2.3275 | 75.326 | 0.9341 |
> | | LO-SDA($\tau=0.0010$) | 0.0481 | 0.113 | 79.1052 | 1.1831 | 1.9254 | 2.2804 | 69.8194 | 0.8965 |
> | | LO-SDA($\tau=0.0001$) | 0.0498 | 0.1188 | 82.0772 | 1.2408 | 1.9602 | 2.3194 | 69.4214 | 1.0308 |

---

> ### Author Response · Authors · 2025-11-27
>
> ___
> > **Response for Q4 & W2:** Performance at Extreme Sparsity and Improvement Magnitude
>
> We thank you for this insightful question. While the performance at 1% sparsity is statistically comparable to 3DVAR (traditional DA) and **L3DVAR (an advanced ML-variational method)**, it is crucial to highlight two points. First, our performance is significantly **superior to other diffusion-based methods** like DPS and Repaint, as shown in our baseline comparisons. This demonstrates the effectiveness of our main contribution—iterative optimization. Second, achieving parity with an advanced method like L3DVAR is, in itself, a **significant breakthrough that aligns with our central claim**. Our work is the **first to demonstrate that a score-based DA framework can match or even surpass highly optimized variational methods** in a global, high-dimensional setting. This result provides the first strong proof-of-concept that replacing the long-standing Gaussian prior assumption with a learned, expressive generative prior is not only feasible but beneficial. The fact that our advantage becomes decisive at 5% observation density further strengthens this claim, showing the scalability of our approach as more data becomes available.
> ___
> > **Response for W1:** EnKF Baseline Comparisons
>
> We agree that a comparison with an advanced traditional ensemble DA method like EnKF is important. However, this is a substantial undertaking beyond this initial study for two reasons. First, EnKF fundamentally estimates a dynamic Gaussian prior through an ensemble of samples, whereas our work focuses on learning and utilizing a non-Gaussian prior. Moreover, its implementation requires ensemble forecasts from the ML model. However, current ML forecasting models exhibit limited sensitivity to ensemble perturbations, which makes EnKF challenging to realize in practice. In comparison, the 3DVAR is a typical variational-based DA, making it a suitable traditional baseline. Our comparison results have demonstrated that a learned, data-driven prior with strict observation enforcement can yield superiority.
> ___
> > **Response for W4:** Theoretical Limitations
>
> We thank the reviewer for these important questions. It seems there may have been a misunderstanding of our theoretical framework, which we are happy to clarify. Our method is not a single-point estimate based on maximum a posteriori (MAP). In LO-SDA, once the optimization step finds a state that strictly conforms to the observation, the resampling step then **projects this state back onto the learned, non-Gaussian manifold of the background-conditioned prior**. This two-step "optimization-projection" process is the generative analogue of the classical Bayesian update, which naturally **balances the information from observation with the constraint from the prior**. The final analysis is therefore not just a point that fits the data, but one that is also highly plausible under our powerful, learned prior. This balancing act is fundamental to all data assimilation and is a core principle of our design.
> ___
> > **Response for W5:** Computational Cost
>
> We acknowledge that the runtime is a key consideration. In response to your feedback, we conducted a new convergence analysis (now in the Appendix). The results show that **nearly identical performance can be achieved with significantly fewer sampling steps and optimization iterations**, reducing the analysis time from ~5 minutes to approximately **1 minute**. This finding dramatically improves the practical outlook of our method. We will clarify it in the paper.
> ___
> > **Response for W6:** Pretraining Assumptions and Generalization
>
> Thank you for this insightful comment. Our VAE is trained on ERA5, a fully opened reanalysis dataset, which serves as the best available estimate of the true atmospheric state. This is a standard and accepted practice for building foundational models in the ML-for-weather community, ensuring that our VAE learns a latent space that is representative of realistic atmospheric dynamics. Moreover, our approach of incorporating observations during the reverse sampling process is a standard and highly efficient practice in the field of conditional generative modeling. A key strength of this approach is its **flexibility and zero-shot generalization capability**. Because the observation operator H is integrated at inference time, LO-SDA can, in principle, assimilate **any type of observation** for which a differentiable operator can be defined, **without any retraining of the base model**. This is a significant advantage over methods that might require end-to-end training for specific observation types. We have also demonstrated the stability of LO-SDA with year-long cycle DA and generalizability with real-world observation (Table 5).
> ___
> We sincerely hope that our responses can address your concerns. Your feedback is valuable in helping us improve the clarity, rigor, and positioning of our work. We welcome any further questions.

---

### Official Review · Reviewer_E3RS · 2025-10-31

**Soundness:** 3
**Presentation:** 3
**Contribution:** 3
**Rating:** 6
**Confidence:** 2

**Summary:**

This paper introduces LO-SDA, a generative data assimilation (DA) framework that replaces traditional Gaussian priors with a latent diffusion model conditioned on background atmospheric states. By incorporating an iterative latent optimization step during the diffusion reverse process, the method ensures strict consistency with sparse observations. Experiments on global ERA5 datasets show that LO-SDA outperforms both traditional variational DA (3DVAR) and its latent variant (L3DVAR), particularly under sparse observation settings.

**Strengths:**

Establishes a theoretical bridge between variational DA and score-based diffusion modeling.

**Weaknesses:**

- Computational cost for latent optimization process is expensive
- The work mainly integrates known elements (VAE compression + score-based DA + optimization) rather than proposing fundamentally new generative principles.

**Questions:**

1. Can the computational cost be reduced in the futher?
2. The paper notes a significant computational burden. Could the authors clarify which parts of the process dominate runtime (diffusion sampling, latent optimization, or decoding)?
3. The ERA5 experiments are convincing, but how does LO-SDA scale to operational forecast cycles or real-time assimilation scenarios?

---

> ### Author Response · Authors · 2025-11-27
>
> We sincerely thank the reviewer for their valuable time and insightful feedback. We are particularly encouraged that you recognized our effort in **establishing a theoretical bridge between variational** **DA** **and score-based diffusion modeling**, a key contribution of our paper. We have carefully considered your comments and provide detailed responses below.
>
>
> ___
>
> > **Response for Question** **1 and Weakness 1:** Computational cost
>
> We agree that the computational cost is a significant consideration. Our initial ~5-minute runtime represented a proof-of-concept where we prioritized SOTA accuracy over efficiency. However, inspired by reviewer feedback, we conducted a new convergence analysis (below table). It reveals that nearly identical performance can be achieved with far fewer sampling steps and optimization iterations, implying that cost is a solvable engineering challenge rather than a fundamental limitation.
>
> | Sampling Steps (with 100 iterations) | DA-MSE (Mean ± Std) |      | Iterations (with 128 sampling steps) | DA-MSE (Mean ± Std) |
> | :----------------------------------: | :-----------------: | :--: | :----------------------------------: | :-----------------: |
> |                  4                   |   0.0537 ± 0.0037   |      |                  1                   |   0.0573 ± 0.0042   |
> |                  8                   |   0.0409 ± 0.0035   |      |                  5                   |   0.0505 ± 0.0039   |
> |                  16                  |   0.0354 ± 0.0035   |      |                  10                  |   0.0444 ± 0.0037   |
> |                  32                  |   0.0334 ± 0.0031   |      |                  20                  |   0.0363 ± 0.0035   |
> |                  64                  | **0.0325 ± 0.0030** |      |                  30                  |   0.0332 ± 0.0030   |
> |                  96                  | **0.0321 ± 0.0027** |      |                  40                  |   0.0324 ± 0.0026   |
> |                 128                  |   0.0326 ± 0.0028   |      |                  50                  | **0.0323 ± 0.0024** |
> |                 192                  |   0.0325 ± 0.0026   |      |                  80                  | **0.0325 ± 0.0024** |
> |                 256                  |   0.0329 ± 0.0025   |      |                 100                  |   0.0326 ± 0.0028   |
> |                  -                   |          -          |      |                 150                  |   0.0328 ± 0.0030   |
> |                  -                   |          -          |      |                 200                  |   0.0327 ± 0.0031   |
>
>
>
> Our new finding demonstrates that the runtime can be reduced to approximately **1 minute per analysis** without a significant drop in accuracy. This strengthens the practical scalability of our method. This runtime is competitive to highly-optimized 3DVAR (~10s on GPU, ~40min on CPU). We will update the manuscript to reflect this improved efficiency and discuss further optimization paths (e.g., model distillation, efficient samplers, multi-GPU paralleling).
>
>
> ___
>
> > **Response for Question 2:** Runtime in different parts
>
> Thank you for the question. In the table below, we present the details about the runtime with 64 diffusion sampling steps, where 25% applied latent optimization using 50 iterations. The bottleneck is explicitly the iterative optimization, which consumes **~92%** of the total runtime. While standard diffusion sampling is fast (0.15s/step), the rigorous constraint enforcement requires multiple latent updates (50 iterations) at specific steps, which accumulates the majority of the computational cost.
>
> | Component                    | Configuration                             |  Time Cost   |   Ratio   |
> | :--------------------------- | :---------------------------------------- | :----------: | :-------: |
> | **Pure Diffusion Sampling**  | 48 steps $\times$ 0.15s                   |    7.2 s     |   8.3%    |
> | **Latent Optimization (LO)** | 16 steps $\times$ 50 iters $\times$ 0.10s |  **80.0 s**  | **91.7%** |
> | **VAE Encode/Decode**        | Included in above                         | *Negligible* |  < 0.1%   |
> | **Total Runtime**            | -                                         |  **87.2 s**  |   100%    |

---

> ### Author Response · Authors · 2025-11-27
>
> ___
>
> > **Response for Question 3:** Operational forecast cycles or Real-time assimilation scenarios
>
> This is a very critical question for practical application. To evaluate the long-term stability required for operational cycles, we conducted a new, extensive **year-long cyclical data assimilation experiment** with 5% observation and every 48h assimilation for 2019. The results, shown in the following table (now in the Appendix), show that analysis RMSE for all key variables remains stable and consistently low throughout the entire year, exhibiting **no signs of error accumulation or** **divergence**. This provides strong empirical evidence that our framework is robust enough for continuous, operational-like sequential assimilation.
>
>
>
> | Var | Mean RMSE (Bg -> **Ours**) | Std Dev (Bg -> **Ours**) | Max Peak (Bg -> **Ours**) | Imp. |
> | :--- | :---: | :---: | :---: | :---: |
> | **z500** | 147.7 $\to$ **55.6** | 18.5 $\to$ **5.7** | 205.7 $\to$ **76.3** | **62.4%** |
> | **u500** | 3.12 $\to$ **2.12** | 0.17 $\to$ **0.09** | 3.58 $\to$ **2.39** | **32.0%** |
> | **t500** | 0.96 $\to$ **0.70** | 0.07 $\to$ **0.03** | 1.13 $\to$ **0.80** | **27.3%** |
> | **v500** | 3.17 $\to$ **2.07** | 0.19 $\to$ **0.10** | 3.67 $\to$ **2.34** | **34.7%** |
> | **u10** | 1.63 $\to$ **1.08** | 0.11 $\to$ **0.06** | 1.96 $\to$ **1.30** | **33.5%** |
> | **v10** | 1.68 $\to$ **1.10** | 0.11 $\to$ **0.06** | 2.06 $\to$ **1.36** | **34.2%** |
> | **t2m** | 1.12 $\to$ **1.04** | 0.07 $\to$ **0.04** | 1.32 $\to$ **1.16** | **6.5%** |
> | **msl** | 154.96 $\to$ **63.85** | 19.70 $\to$ **8.28** | 223.28 $\to$ **109.32** | **58.8%** |
>
> Regarding computational cost, our method is competitive to the operational 3DVAR baseline (40 minutes on a single CPU core) with a runtime of just 1 minute on GPU. Moreover, by leveraging established diffusion sampling acceleration techniques, it is promising to reduce this cost to seconds, moving towards the goal of real-time assimilation.
>
>
>
>
> ___
>
> > **Response for Weakness 2**: Novelty of the Proposed Integrated Framework
>
> We thank the reviewer for this thoughtful comment. We acknowledge that our framework integrates existing architectures (VAE, diffusion, and optimization). However, we respectfully clarify that our primary contribution is not proposing a new generative principle, but rather identifying and overcoming the fundamental barriers that have prevented generative models from surpassing traditional DA.
>
> **First, we address a theoretical bottleneck.** Traditional DA methods (e.g., 3DVAR) are fundamentally limited by Gaussian prior assumptions. While the community has hypothesized that deep generative models could offer superior non-Gaussian priors, this had not been convincingly proven against strong baselines. Our work validates this hypothesis, demonstrating that replacing the Gaussian assumption with a learned diffusion prior yields theoretically superior state estimation.
>
> **Second, we demonstrate that realizing this potential is non-trivial.** As noted, standard integrations (e.g., DPS) fail to outperform traditional DA in high-dimensional settings. Our novelty lies in the specific algorithmic design of LO-SDA, which shifts the paradigm from the "single-step gradient guidance" of DPS to "iterative latent optimization." This design mimics the rigorous constraint enforcement of Variational DA within the generative process.
>
> The effectiveness of this integrated framework is evidenced by our results: a **13.39% improvement over the DPS baseline** and superior performance to the 3DVAR. This provides the first strong proof-of-concept that generative DA can practically outperform traditional methods, transforming "known elements" into a state-of-the-art solution.
>
>
>
> ___
>
> We sincerely hope that these clarifications and the new experimental results have addressed your concerns. We are grateful for your constructive feedback, which has helped us to significantly improve the paper. We welcome any further questions or suggestions you may have.

---

> > ### Comment · Reviewer_E3RS · 2025-11-27
> >
> > I thank author for their thoughtful response, and additional explanation and effort. Overall, I am happy with the result, and I would like to retain my score as it is now.

---

> > > ### Author Response · Authors · 2025-11-27
> > >
> > > We sincerely thank the reviewer for the positive feedback and for recognizing our efforts. We are glad to hear that our response and additional experimental results have addressed your concerns. If there are any further questions, feel free to raise them, and we are always ready for the discussion.
> > >
> > > Best regards,
> > >
> > > The Authors

---

### Official Review · Reviewer_fudT · 2025-11-01

**Soundness:** 3
**Presentation:** 3
**Contribution:** 3
**Rating:** 6
**Confidence:** 3

**Summary:**

The paper proposes LO-SDA, a score-based data assimilation (DA) framework that integrates latent optimisation with background-conditioned diffusion models. A variational autoencoder (VAE) is used to learn low-dimensional latent representations of atmospheric states; within this space, a diffusion model learns the conditional prior. During reverse diffusion, the authors introduce latent optimisation steps that iteratively enforce observation consistency, which they interpret as a generative analogue of variational cost-function minimisation. Experiments on ERA5 global reanalysis show that LO-SDA achieves performance comparable to or slightly better than variational DA (L3DVAR), outperforming single-step diffusion-based methods such as DPS and Repaint under sparse-observation settings.

**Strengths:**

Clear motivation. The work targets the Gaussian-prior limitation of classical DA and builds a principled bridge between score-based and variational formulations.

Methodological coherence. The latent optimisation scheme is well justified and neatly integrated into the diffusion framework. Algorithm 1 clearly contrasts it with DPS guidance.

Presentation quality. Figures are clear and the connection to variational DA is articulated.

**Weaknesses:**

- Incremental novelty. The paper’s central idea—enforcing observation consistency within score-based DA—is conceptually close to existing approaches such as score-based DA, DiffDA, and Manshausen et al., 2025. LO-SDA differs mainly by performing optimization in latent space rather than directly in model space and by framing the repeated guidance as “latent optimization.” This is a useful but evolutionary step rather than a paradigm shift.

- Overstated claims. The manuscript repeatedly suggests LO-SDA is the first score-based method to outperform traditional DA, but the margin over L3DVAR is modest and depends on selected metrics and observation density. The contribution would be stronger if positioned as a refinement of existing score-based methods rather than a breakthrough.

- Computational cost. Iterative latent optimisation requires ~5 minutes per assimilation versus seconds for variational baselines; scalability to operational use remains unclear.

- Limited theoretical depth. The link between latent optimisation and variational minimisation is described heuristically without formal guarantees or convergence analysis.

- Related-work positioning. Discussion could better situate LO-SDA among other generative and variational DA frameworks, e.g., Tensor-Var (2025) or DiffDA (2024), highlighting specific differences in assumptions and efficiency instead of focusing primarily on 3DVAR/L3DVAR comparisons.

**Questions:**

- Does the optimization always converge, and how many iterations are typically needed?
- How does LO-SDA handle strongly nonlinear observation operators such as satellite radiances?

---

> ### Author Response · Authors · 2025-11-27
>
> We sincerely thank the reviewer for the thoughtful and constructive feedback.
>
>
> ___
>
> > **Response for Question** **1 and Weakness 4:** Theoretical Depth and Convergence
>
> We acknowledge that establishing a **formal, mathematically rigorous equivalence** between latent optimization and variational minimization is a significant challenge, complicated by the non-convex latent space and the stochastic nature of the diffusion process. In addition, the resampling step lacks a closed-form analytical expression comparable to the quadratic background term. However, existing studies show that the decoder of geoscience data–trained VAEs is locally near-linear, which implies that optimization in the latent space does not alter convexity [1,2]. This property is a key reason why latent-space assimilation methods remain effective.
>
> While a formal proof of global convergence is challenging (a common open problem), we provide **empirical evidence of convergence**. Motivated by your feedback, we conducted new ablation studies (the table below) on the number of diffusion sampling steps and latent optimization iterations. The analysis shows that the DA-MSE converges and stabilizes after approximately 32 sampling steps and 50 optimization iterations. This not only confirms that the hyperparameters used in our main experiments (128 sampling steps, 100 optimization iterations) were in the converged regime, but also demonstrates that our algorithm is stable, well-performed, ensuring our theoretical analogy is realized through a robust and effective algorithm in practice.
>
> | Sampling Steps (with 100 iterations) | DA-MSE (Mean ± Std) |      | Iterations (with 128 sampling steps) | DA-MSE (Mean ± Std) |
> | :----------------------------------: | :-----------------: | :--: | :----------------------------------: | :-----------------: |
> |                  4                   |   0.0537 ± 0.0037   |      |                  1                   |   0.0573 ± 0.0042   |
> |                  8                   |   0.0409 ± 0.0035   |      |                  5                   |   0.0505 ± 0.0039   |
> |                  16                  |   0.0354 ± 0.0035   |      |                  10                  |   0.0444 ± 0.0037   |
> |                  32                  |   0.0334 ± 0.0031   |      |                  20                  |   0.0363 ± 0.0035   |
> |                  64                  | **0.0325 ± 0.0030** |      |                  30                  |   0.0332 ± 0.0030   |
> |                  96                  | **0.0321 ± 0.0027** |      |                  40                  |   0.0324 ± 0.0026   |
> |                 128                  |   0.0326 ± 0.0028   |      |                  50                  | **0.0323 ± 0.0024** |
> |                 192                  |   0.0325 ± 0.0026   |      |                  80                  | **0.0325 ± 0.0024** |
> |                 256                  |   0.0329 ± 0.0025   |      |                 100                  |   0.0326 ± 0.0028   |
> |                  -                   |          -          |      |                 150                  |   0.0328 ± 0.0030   |
> |                  -                   |          -          |      |                 200                  |   0.0327 ± 0.0031   |
>
> [1] https://arxiv.org/abs/2502.02884
>
> [2] https://arxiv.org/abs/2510.04006
>
>
>
>
> ___
>
> > **Response for Question 2:** Nonlinear Observation Operators
>
> Thanks for raising this important point. LO-SDA can handle nonlinear operators naturally, as long as 𝐻 is implemented in a differentiable form within the deep learning framework. Specifically, the optimization step `∇_z ||y - H(D(z))||^2` handles any differentiable operator H automatically through the chain rule via automatic differentiation (AD).  We note that this property is promising, considering that many recent studies implement observation operators within deep learning frameworks or even replace them entirely with neural-network-based observation operators.

---

> ### Author Response · Authors · 2025-11-27
>
> ___
>
> > **Response for Weakness 1:** Novelty of LO-SDA
>
> We would like to respectfully clarify that LO-SDA represents a significant conceptual and empirical contribution, not an incremental step. Although many studies have explored SDA, to the best of our knowledge, none have demonstrated whether it can surpass traditional data assimilation, nor explained why it would be able to do so.
>
> The first contribution of this paper is theoretical. We point out that traditional data assimilation—also based on Bayesian principles—suffers from a fundamental limitation: the prior is constrained to be Gaussian. By contrast, a diffusion model can learn a much more expressive prior and therefore has the potential to surpass traditional DA. The second contribution is methodological. We show that most existing SDA[2,3,4] approaches incorporate observational information through the DPS mechanism, but this strategy is unable, in practice, to outperform traditional DA.
>
> This is where our key innovation comes in. LO-SDA moves from "single-step gradient guidance" to **"iterative latent optimization**. This shift is designed to mimic the rigorous constraint enforcement that is the key to variational DA's success. The impact of this methodological innovation is a clear performance breakthrough. As shown in our results, LO-SDA significantly outperforms the DPS baseline by **13.39% in** **MSE** (with 1% observations). Crucially, it is the **first generative** **DA** **framework to achieve performance comparable to or superior to L3DVAR**. This result serves as a **proof-of-concept** for a foundational hypothesis in our field: that relaxing the Gaussian background assumption with a learned, data-driven prior can yield superior results. By providing this first piece of strong evidence, we validate the potential for generative models to complement traditional DA. We contend this validation constitutes the core, non-incremental novelty of our work.
>
> [2] https://arxiv.org/abs/2306.10574
>
> [3] https://arxiv.org/abs/2401.05932
>
> [4] https://arxiv.org/abs/2406.16947v2
>
>
> ___
>
> > **Response for Weakness 2:** Overstated Claims
>
> We thank the reviewer and will revise the manuscript to position our claims more carefully. We note that L3DVar is not a traditional DA method, but rather a new advanced framework that combines ML with traditional DA. The truly traditional data assimilation baseline is 3DVar, and LO-SDA shows a clear and substantial advantage over it.
>
> We note that although L3DVar adopts a Gaussian prior in the latent space, it becomes dynamic in the model space as it evolves with the background field. In contrast, 3DVar relies on a static prior in the model space.
>
>  Therefore, although both assume Gaussianity, the flow-dependent nature of L3DVar provides a substantial improvement over 3DVar. Our method further achieves a modest but consistent advantage over L3DVar, highlighting the benefit of replacing the Gaussian prior with a diffusion-based prior.
>
> Therefore, we believe our claims are appropriately supported. We have emphasized these points clearly in the main text.
>
>
> ___
>
> > **Response for Weakness 3:** Computational Cost
>
> We thank the reviewer for raising this critical point. We acknowledge our current runtime (~5 min/GPU) is higher than the highly optimized 3DVAR (~10s on GPU, ~40min on CPU). Our goal was a **proof-of-concept**, accepting a computational trade-off to first establish SOTA accuracy. Importantly, our new **convergence** **analysis** (in Appendix) revealed that we can achieve nearly identical performance with far fewer sampling steps and iterations. It is possible to **reduce the runtime to approximately 1 minute per analysis**. This finding significantly strengthens the scalability of our method. We will update the manuscript to reflect this improved efficiency and to discuss further optimization paths (e.g.,  **model distillation, efficient samplers, and Multi-GPU paralleling**), confirming that this is an engineering challenge with clear solutions, not a fundamental limitation.
>
>
> ___
>
> > **Response for Weakness 5:** Related-work Positioning
>
> Thank you for this suggestion. We will revise Section 2 to clarify our positioning. Unlike **DiffDA**'s single-step guidance, our **"iterative latent optimization** enforces a much stricter data constraint. While **Tensor-Var** improves the variational framework, LO-SDA is a generative-native approach. Our unique contribution is the **hybrid integration of a variational-like optimization directly into the diffusion sampling process**, which closes the performance gap with SOTA DA methods.
>
>
> ___
>
> We sincerely appreciate your valuable feedback, which has significantly enhanced the clarity and completeness of our paper.

---

### Author Response · Authors · 2025-11-28
**General Response**

Dear Reviewers and Area Chairs,

We sincerely thank you for your thorough, insightful, and constructive reviews.

___

We are greatly encouraged that the reviewers recognize the novelty, theoretical grounding, and potential impact of our work:

*   **Reviewer fudT** highlighted the work’s "clear motivation" and "methodological coherence," noting that the latent optimization scheme is "well justified and neatly integrated."
*   **Reviewer E3RS** acknowledged that LO-SDA effectively "establishes a theoretical bridge between variational DA and score-based diffusion modeling."
*   **Reviewer kg13** recognized the work as a "creative combination" and a "potential paradigm shift" that challenges the long-standing reliance on Gaussian assumptions.
*   **Reviewer ZLd6** noted the method's "respectable improvement".

___

In response to your valuable feedback, we have performed additional experiments and revisions to strengthen the manuscript::

*   **Theoretical Depth and Convergence (Addressing fudT, kg13):**
    We conceptualize LO-SDA as the generative analogue to variational DA, balancing the diffusion prior with rigorous observation enforcement. While formal equivalence is challenging due to non-convexity, we provide **strong empirical evidence of convergence**. New ablation studies confirm that the optimization stabilizes reliably, ensuring the algorithm is robust in practice.

*   **Computational Efficiency (Addressing fudT, E3RS, kg13):**
    The **new convergence analysis** demonstrates that LO-SDA achieves nearly identical performance with significantly fewer sampling steps and optimization iterations. It is promising to reduce the inference time to **1 minute per analysis**, making it competitive with traditional baselines and proving that efficiency is an engineering  problem rather than a fundamental limitation.

*   **Long-Term Sequential Stability (Addressing kg13, ZLd6):**
    To address concerns regarding error accumulation, we performed a **new year-long cyclical data assimilation experiment** (2019, every 48h, 5% observation). The results demonstrate that the analysis RMSE remains stable and consistently low throughout the entire year with **no signs of divergence**, providing strong empirical evidence that LO-SDA is robust for cyclic assimilation.

*   **Robustness to High Noise (Addressing kg13, ZLd6):**
    We conducted a new experiment under high-noise conditions ($\text{std}=0.1$) by relaxing the optimization constraint ($\tau$). The results show that LO-SDA can flexibly balance the diffusion prior with observation enforcement, achieving superior performance even with noisy data.

*   **Clarification of Novelty & Positioning (Addressing fudT, E3RS):**
    We have revised the manuscript to explicitly distinguish LO-SDA through two key contributions:
    1.  **Overcoming the Gaussian Barrier:** We provide the first strong proof-of-concept that a learned, non-Gaussian diffusion prior can surpass traditional Gaussian-based methods (3DVar/L3DVar) in high-dimensional settings.
    2.  **From "Guidance" to "Optimization":** Unlike existing score-based methods (e.g., DPS) that rely on weak "single-step guidance," we introduce **"Iterative Latent Optimization."** This methodological shift strictly enforces observational constraints, enabling LO-SDA to significantly outperform diffusion baselines and match advanced variational methods.

___

We believe these new experiments and clarifications directly address the concerns raised. LO-SDA provides the first strong proof-of-concept that a learned, non-Gaussian generative prior—when rigorously constrained via optimization—can surpass traditional variational DA.

We welcome any further feedback and look forward to the discussion.

Best regards,

The Authors

---

### Meta-Review · Area_Chair_DRCP · 2026-01-01

**Summary:**

The paper proposes LO-SDA, a background-conditioned latent diffusion prior coupled with iterative latent optimization to enforce observation consistency, and the reviewers generally agree that the problem is well-motivated and the method is coherent, with convincing results that outperform single-step diffusion guidance baselines and are competitive with existing methods under certain sparse-observation settings. The rebuttal adds more useful evidence—an ablation-style convergence/runtime analysis suggesting the method can be sped up, and additional experiments on long-horizon cycling and high-noise regimes. These results partially addressed concerns about cost, stability, and robustness.

However, the key reservations remain substantial for acceptance: (i) the novelty is viewed by multiple reviewers as incremental (a combination of VAE compression + score-based DA + repeated optimization) with positioning and “first/breakthrough” claims still feeling overstated relative to the modest margins and regime dependence; (ii) the empirical evaluation is incomplete with respect to major state-of-practice baselines and closely related latent/score-based DA lines (e.g., ensemble-based DA and recent latent-space DA variants); (iii) the method remains largely heuristic (limited guarantees on convergence/optimality, and unclear implications for Bayesian posterior fidelity/uncertainty), and several practical questions about method such as observation operators, end-to-end cycling protocol, and scalability are not fully resolved despite added appendix results.

 The work is promising as a proof-of-concept and would be significantly strengthened by tighter claims, a more comprehensive related-work/baseline suite, and clearer evidence of robustness and practicality beyond the current experimental envelope.

**Reviewer Concerns:**

* Novelty/positioning and completeness of baselines

Concern: Several reviewers felt LO-SDA is an incremental integration of known components (VAE latent space + score-based DA + iterative enforcement) and that “paradigm shift/first” claims are overstated, especially given modest or regime-dependent gains and missing comparisons to key DA/latent-DA baselines.

How authors addressed: Reframed the core novelty as moving from single-step DPS-style guidance to iterative latent optimization that more strictly enforces observations, promising stronger related-work positioning.

* Computational cost and operational scalability

Concern: The inference cost and optimization overhead raised doubts about real-time applicability.

How authors addressed: Added an ablation/convergence study showing near-identical performance with fewer diffusion steps/optimization iterations and claimed runtime can be reduced to ~1 minute; provided a runtime breakdown indicating optimization is the bottleneck and argued this is an engineering/acceleration problem.

* Theoretical rigor and sequential robustness

Concern: The variational-analogy is largely heuristic, and reviewers questioned whether repeated/cyclic assimilation is stable (error accumulation), especially under noise.

The authors added empirical convergence evidence (DA-MSE stabilizing after certain steps/iters), a year-long cyclic DA experiment reporting stable RMSE without divergence, and high-noise tests showing robustness can be tuned via the optimization strength/threshold.

**Reviewer Scores:**

none

---

### Decision · Program_Chairs · 2026-01-26

Reject